# Transformers Learn Low Sensitivity Functions: Investigations and Implications

**Bhavya Vasudeva[†], Deqing Fu[†], Tianyi Zhou, Elliott Kau[\*], Youqi Huang[\*], Vatsal Sharan**
University of Southern California
`bvasudev@usc.edu, deqingfu@usc.edu`

## Abstract

Transformers achieve state-of-the-art accuracy and robustness across many tasks, but an understanding of their inductive biases and how those biases differ from other neural network architectures remains elusive. In this work, we identify the sensitivity of the model to token-wise random perturbations in the input as a unified metric which explains the inductive bias of transformers across different data modalities and distinguishes them from other architectures. We show that transformers have lower sensitivity[1] than MLPs, CNNs, ConvMixers and LSTMs, across both vision and language tasks. We also show that this low-sensitivity bias has important implications: i) lower sensitivity correlates with improved robustness; it can also be used as an efficient intervention to further improve the robustness of transformers; ii) it corresponds to flatter minima in the loss landscape; and iii) it can serve as a progress measure for grokking. We support these findings with theoretical results showing (weak) spectral bias of transformers in the NTK regime, and improved robustness due to the lower sensitivity.

## 1 Introduction

Transformers, originally introduced for language problems (Vaswani et al., 2017), have become a universal backbone across machine learning — including applications such as vision (Dosovitskiy et al., 2021) and protein structure prediction (Jumper et al., 2021). Several recent works have also found that not only do transformers achieve better accuracy, but they are also more robust to various corruptions and changes in the data distribution (Shao et al., 2021; Mahmood et al., 2021; Bhojanapalli et al., 2021; Paul & Chen, 2022). Despite their practical success, relatively little is understood about what distinguishes transformers from other neural network architectures. *If a transformer and an alternative neural network architecture (such as a CNN or an LSTM) are trained to obtain similar training accuracy on a dataset, then how do the models differ in terms of the functions they learn? Equivalently, what inductive biases do transformers have which distinguish them from other architectures?*

Recently, for the setting of Boolean inputs, Bhattamishra et al. (2023b) and Hahn & Rofin (2024) suggest using the notion of *sensitivity* to distinguish transformers from other candidate architectures. The sensitivity of a function measures how likely the output is to change for random changes to the input. Bhattamishra et al. (2023b) and Hahn & Rofin (2024) show that transformers are biased to learn functions with low sensitivity on Boolean inputs. Sensitivity has several desirable properties as a notion of inductive bias. It is closely related to the Fourier representation of the function and various other notions of Boolean function complexity such as the degree of the function and the size of the smallest decision tree which represents the function (O'Donnell, 2014). Low sensitivity functions correspond to low complexity functions based on all these notions of Boolean function complexity, and hence an inductive bias towards low-sensitivity functions is regarded as an instance of 'simplicity bias' of the model (Valle-Perez et al., 2019; Bhattamishra et al., 2023b). Sensitivity also has deep connections to well-studied notions of inductive biases such as spectral bias (Rahaman et al., 2019b), which is a bias towards 'simple' functions such as low frequency functions in the Fourier space. Sensitivity has also been found to correlate with better generalization for fully-connected networks (Novak et al., 2018). (See Appendix C for a detailed discussion of the related work.)

---

[†]Co-first authors. [\*]Co-third authors. [1]The code is available at `https://github.com/estija/sensitivity`.

**Our results.** Sensitivity is a promising notion of inductive bias, but has mainly been investigated for Boolean functions so far. Given the numerous modalities of data across which transformers are successful in practice, the goal of our work is to examine if appropriate extensions of the notion of sensitivity for Boolean functions help understand the inductive bias of transformers across varied data modalities — and if these notions help explain properties of transformers such as their improved robustness. We now provide an overview of the main claims and results of the paper. We begin our investigation with the following question:

*What are appropriate notions of sensitivity beyond Boolean data?*

To provide a concrete starting point where we can understand the properties of sensitivity with theoretical analysis, we first consider the Boolean setup and place the low-sensitivity bias of transformers on a firmer theoretical foundation in that setting (Section 2). Using prior work on spectral bias in neural networks (Yang & Salman, 2020) we prove that transformers show a low-sensitivity bias on Boolean functions, and also prove that low sensitivity leads to better robustness. We then consider the above question, and propose a suitable notion of sensitivity which takes into account the underlying metric space (Section 3). To investigate if sensitivity is a suitable notion beyond the Boolean case, we first consider a synthetic dataset where we can tease apart sensitivity from related notions which can coincide for Boolean functions — such as a preference towards functions that depend on a sparse set of tokens. We show that transformers prefer to learn low-sensitivity functions (even if they are not sparse). Subsequently, we examine if this low-sensitivity bias is widely present across different tasks:

*Does low-sensitivity serve as a unified notion of simplicity across vision and language tasks, and does it distinguish between transformers and other architectures?*

Here, we first conduct experiments on vision datasets. We empirically compare (Vision-)Transformers with MLPs, CNNs, and ConvMixers, and observe that transformers have lower sensitivity compared to other candidate architectures (see Section 4). Similarly, we conduct experiments on language tasks and observe that transformers learn predictors with lower sensitivity than LSTM models. Furthermore, transformers tend to have uniform sensitivity to all tokens while LSTMs are more sensitive to more recent tokens (see Section 5). Given this low-sensitivity bias, we next examine its implications:

*What are the implications of a bias towards low-sensitivity functions; is it helpful in certain settings?*

We study this in three contexts: robustness, properties of the loss landscape, and training dynamics.

1. **Lower Sensitivity Correlates with Better Robustness**: We show that transformers have lower sensitivity and are more robust to corruptions when tested on the CIFAR-10-C dataset, compared to CNNs (Section 6). We also demonstrate that sensitivity is not only predictive of robustness but also has prescriptive power: We add a regularization term at training time to encourage the model to have lower sensitivity. Since sensitivity is efficient to measure empirically, this is easy to accomplish via data augmentation. We find that models explicitly trained to have lower sensitivity yield even better robustness on CIFAR-10-C. These results show that *low sensitivity correlates with the improved robustness of transformers*.
2. **Lower Sensitivity Correlates with Flatter Minima**: We explore the connection between sensitivity and a property of the loss landscape that has been found to correlate to good generalization — the sharpness of the minima. We compare the sharpness of the minima with and without sensitivity regularization, and our results show that *lower sensitivity correlates with flatter minima*. This indicates that sensitivity could serve as a unified notion for both robustness and generalization.
3. **Sensitivity Serves as a Progress Measure for Grokking**: We examine if sensitivity can be used to understand the training dynamics of transformers, specifically from the perspective of the phenomenon of *grokking* where test accuracy abruptly improves long after the training loss or accuracy saturates. We consider modular addition, a task where transformers exhibit grokking. We show that *sensitivity provides a progress measure that decreases even when the training loss does not reduce and is indicative of stages of grokking*.

## 2   SENSITIVITY AND WEAK SPECTRAL BIAS

In this section, we theoretically show that transformers with linear attention exhibit (weak) spectral bias to learn lower-order Fourier coefficients, which in turn implies a bias to learn low-sensitivity functions. We start with an overview of Fourier analysis on the Boolean cube and sensitivity.

**Fourier analysis on the Boolean cube (O'Donnell, 2014).** The space of real-valued functions on the Boolean cube $\boxplus^d$ forms a $2^d$-dimensional space. Any such function can be written as a *unique multilinear* polynomial. Specifically, the multilinear monomial functions, $\chi_U(\boldsymbol{x}) := x^U := \prod_{i \in U} x_i$, for each $U \subseteq [d]$, form a Fourier basis of the function space $\{f : \boxplus^d \to \mathbb{R}\}$, *i.e.*, their inner products satisfy $\mathbb{E}_{\boldsymbol{x} \sim \boxplus^d}[\chi_U(\boldsymbol{x})\chi_V(\boldsymbol{x})] = \mathbb{1}[U = V]$. Consequently, any function $f : \boxplus^d \to \mathbb{R}$ can be written as $f(\boldsymbol{x}) = \sum_{U \subseteq [d]} \hat{f}(U)\chi_U(\boldsymbol{x})$, for a unique set of coefficients $\hat{f}(U), U \subseteq [d]$, where $[d] = \{1, \ldots, d\}$.

**Sensitivity in Boolean function analysis.** Sensitivity is a common complexity measure for Boolean functions. Intuitively, it captures the changes in the output of the function, averaged over the neighbours of a particular input. Formally, let $\boxplus^d := \{\pm 1\}^d$ denote the Boolean cube in dimension $d$. The sensitivity of a Boolean function $f : \boxplus^d \to \{\pm 1\}$ at input $\boldsymbol{x} \in \boxplus^d$ is given by $S(f, \boldsymbol{x}) = \sum_{i=1}^d \mathbb{1}[f(\boldsymbol{x}) \neq f(\boldsymbol{x}^{\oplus i})]$, where $\mathbb{1}[\cdot]$ denotes the indicator function and $\boldsymbol{x}^{\oplus i} = (x_1, \ldots, x_{i-1}, -x_i, x_{i+1}, \ldots, x_d)$ denotes the sequence obtained after flipping the $i^{\text{th}}$ coordinate of $\boldsymbol{x}$. Note that in the Boolean case, the neighbor of an input can be obtained by flipping a bit, we will define a more general notion later which holds for more complex data. The average sensitivity is measured by averaging $S(f, \boldsymbol{x})$ across all inputs,

$$S(f) = \mathbb{E}_{\boldsymbol{x} \sim \boxplus^d}[S(f, \boldsymbol{x})] = \frac{1}{2^d} \sum_{\boldsymbol{x} \in \boxplus^d} S(f, \boldsymbol{x}). \tag{1}$$

Following Bhattamishra et al. (2023b), when comparing inputs of different lengths, we consider the average sensitivity normalized by the input length, $\overline{S}(f) = \frac{1}{d}S(f)$. The sensitivity of a function $f$ is known to be related to the degree $D(f)$ of the multilinear polynomial which represents $f$ (Huang, 2019; Hatami et al., 2011), and low-degree functions have lower sensitivity. Specifically, a breakthrough result (Huang, 2019) showed that $D(f) \leq S_{\max}^2(f)$, where $S_{\max}(f) := \max_{\boldsymbol{x} \in \boxplus^d} S(f, \boldsymbol{x})$.

**Attention Layer.** The output of a single-head self-attention layer, parameterized by key, query, value matrices $\boldsymbol{W}_Q, \boldsymbol{W}_K \in \mathbb{R}^{\tilde{d} \times d_h}, \boldsymbol{W}_V \in \mathbb{R}^{\tilde{d} \times d_v}$ for input $\boldsymbol{X} \in \mathbb{R}^{T \times \tilde{d}}$ with $T$ tokens of $\tilde{d}$ dimension, is $\text{ATTN}(\boldsymbol{X}; \boldsymbol{W}_Q, \boldsymbol{W}_K, \boldsymbol{W}_V) := \boldsymbol{\varphi}(\boldsymbol{X}\boldsymbol{W}_Q\boldsymbol{W}_K^\top\boldsymbol{X}^\top)\boldsymbol{X}\boldsymbol{W}_V$, where $\boldsymbol{\varphi}(\boldsymbol{X}\boldsymbol{W}_Q\boldsymbol{W}_K^\top\boldsymbol{X}^\top) \in \mathbb{R}^{T \times T}$ is the attention map with the softmax map $\boldsymbol{\varphi}(\cdot) : \mathbb{R}^T \to \mathbb{R}^T$ applied row-wise.

**Main Results.** Consider any model with at least one self-attention layer, where $\boldsymbol{X}$ is obtained by reshaping $\boldsymbol{x} \in \boxplus^d$, $d = T\tilde{d}$. Instead of applying the softmax activation, we consider linear attention and apply an identity activation element-wise with a scaling factor of $d^{-1/2}$. The following result shows that the conjugate kernel (CK) or neural tangent kernel (NTK) (see Appendix B for an overview) induced by transformers with linear attention exhibit a weak form of spectral bias, where the eigenvalues do not decrease as the degree of the multi-linear monomials increases, separately for even and odd degrees; see Appendix B for the proof.

**Proposition 2.1.** *Let $K$ be the CK or NTK of a transformer with linear attention on a Boolean cube $\boxplus^d$. For any $\boldsymbol{x}, \boldsymbol{y} \in \boxplus^d$, we can write $K(\boldsymbol{x}, \boldsymbol{y}) = \Psi(\langle \boldsymbol{x}, \boldsymbol{y} \rangle)$ for some univariate function $\Psi : \mathbb{R} \to \mathbb{R}$. Further, for every $U \subseteq [d]$, $\chi_U$ is an eigenfunction of $K$ with eigenvalue*

$$\mu_{|U|} := \mathbb{E}_{\boldsymbol{x} \sim \boxplus^d}\left[x^U K(\boldsymbol{x}, \boldsymbol{1})\right] = \mathbb{E}_{\boldsymbol{x} \sim \boxplus^d}\left[x^U \Psi\left(d^{-1} \sum_i x_i\right)\right],$$

*where $\boldsymbol{1} := (1, \ldots, 1) \in \boxplus^d$, and the eigenvalues $\mu_k$, $k \in [d]$, satisfy*

$$\mu_0 \geq \mu_2 \geq \cdots \geq \mu_{2k} \geq \ldots, \quad \mu_1 \geq \mu_3 \geq \cdots \geq \mu_{2k+1} \geq \ldots.$$

Note that for a given $U$, the eigenvalue $\mu_{|U|}$ only depends on $x^U$ and $\sum_i x_i$ by definition, and hence, it is invariant under any permutation of $[d]$. Larger eigenvalues for lower-order monomials indicate that simpler features are learned faster. Since low sensitivity implies learning low-degree polynomials, Proposition 2.1 also implies a weak form of low sensitivity bias.

We now show a connection between lower sensitivity and better robustness. Given a sample $\boldsymbol{x} \in \boxplus^d$ and some $\rho \in (0, 1)$, consider a noisy sample $\boldsymbol{x}'$, where $x_i' = x_i$ with probability $\rho$ and uniformly random, otherwise. It can be verified that the pair $(\boldsymbol{x}, \boldsymbol{x}')$ has correlation $\rho$. The following result

shows that if $f$ has a lower sensitivity $S_{\max}(f)$, then there is a lower probability $\Pr[f(\boldsymbol{x}) \neq f(\boldsymbol{x}')]$ of inconsistent predictions on the pair $(\boldsymbol{x}, \boldsymbol{x}')_\rho$; see Appendix B for the proof.

**Proposition 2.2.** *Given $\rho$-correlated pair $(\boldsymbol{x}, \boldsymbol{x}')$, where $\rho \in (0,1)$, and function $f : \boxed{\phantom{x}}^d \rightarrow \{\pm 1\}$ with maximum sensitivity $S_{\max}(f)$, $0 \leq \Pr\limits_{(\boldsymbol{x}, \boldsymbol{x}')_\rho}[f(\boldsymbol{x}) \neq f(\boldsymbol{x}')] \leq 0.5(1 - \rho^{(S_{\max}(f))^2})$.*

Together, Propositions 2.1 and 2.2 imply that transformers have low sensitivity and hence, better robustness. In Section 6, we present experimental evidence showing that the low sensitivity of transformers correlates with their improved robustness.

## 3 MEASURING SENSITIVITY BEYOND BOOLEAN DATA

While sensitivity appears to be a promising metric to understand the inductive biases of transformers, it is only defined for Boolean data. In order to investigate the inductive biases in real-world image and language tasks, we need an equivalent metric for high-dimensional, real-valued data.

We define the sensitivity metric for high-dimensional data, which is an analog of Eq. (1) as follows.

**Definition 3.1.** Given a model $\Phi$, dataset $\mathcal{D}$ and distribution $\mathcal{P}$, sensitivity is computed as:

$$\overline{S}(\Phi) = \frac{1}{T} \mathop{\mathbb{E}}_{\substack{\boldsymbol{X} \sim \mathcal{D} \\ \boldsymbol{x} \sim \mathcal{P}}} \left[ \sum_{\tau=1}^{T} \mathbb{1}\left[ \text{SIGN}(\Phi(\boldsymbol{\theta}; \boldsymbol{X})) \neq \text{SIGN}(\Phi(\boldsymbol{\theta}; \boldsymbol{X}^{\oplus \tau})) \right] \right], \tag{2}$$

where $\boldsymbol{X}^{\oplus \tau}$ is obtained by replacing the $\tau^{\text{th}}$ token in $\boldsymbol{X}$ with $\boldsymbol{x}$.

An important consideration here is to define $\mathcal{P}$. While one can replace a token with a randomly selected token to measure sensitivity, this may not ensure that the new token lies in a neighbourhood of the original token. Capturing how the output changes with local perturbations according to the metric of the underlying space is an important aspect of the sensitivity definition for Boolean functions (as discussed in Gopalan et al. (2016), for e.g.), and appropriate extensions of sensitivity beyond Boolean functions should capture this property. For input spaces such as natural images or text embeddings, there is more structure in the tokens, and a randomly selected token can lie far from the original token's neighborhood. Therefore, instead of replacing a token with a random token, we inject small perturbations into the token to evaluate sensitivity. This allows us to control the size of the neighbourhood by selecting the strength of the noise perturbation.

Formally, for each token $\boldsymbol{e}_\tau$ of an input $\boldsymbol{X}$, let $\boldsymbol{x} := \boldsymbol{e}_\tau + \boldsymbol{\xi}$ be a perturbed token, where $\boldsymbol{\xi} \sim \mathcal{N}(\boldsymbol{0}, \sigma^2 \boldsymbol{I})$ is an isotropic Gaussian with variance $\sigma^2$. We measure

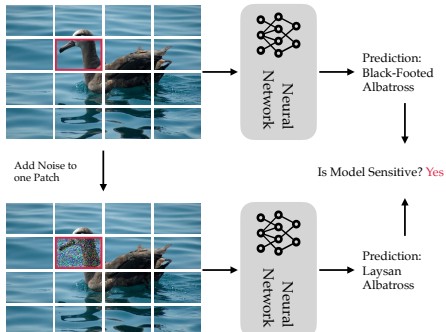

Figure 1: **Measuring Sensitivity in Vision Tasks**. A patch is first selected to add Gaussian noise corruptions. Then the original image and the corrupted image are fed into the *same* neural network to make predictions. If the predictions are inconsistent, then the neural network is sensitive to this patch. The process is repeated for every patch to measure the overall sensitivity.

sensitivity by replacing $\boldsymbol{e}_\tau$ with $\boldsymbol{x}$ as per Definition 3.1, with $\mathcal{P}$ as $\mathcal{N}(\boldsymbol{0}, \sigma^2 \boldsymbol{I})$. For image data, each token $\boldsymbol{e}_\tau$ corresponds to different patches (see Section 4 for further details), while for language data, the tokens correspond to embeddings of sub-words (see Section 5 for more details). Figure 1 illustrates the measurement for images.

The important characteristics of this metric are that it is a unified notion of complexity across vision and language tasks, and as we will see later, it distinguishes transformers from various other architectures. For instance, we compare the sensitivity measured with token-wise perturbations as mentioned above, with random perturbations across the input in Appendix A.3 and find that the gap in the latter metric is not as large as the proposed metric.

We also note that while for Boolean data, sensitivity aligns with other notions of complexity, such as sparsity, this may or may not be the case in settings with high-dimensional or real-valued data. In the following section, we present experimental results for a self-attention model, in a synthetic data setting to demonstrate this. Specifically, we show that in the synthetic dataset, the related notion of

using a sparse set of input tokens may or may not align with low-sensitivity, but the model learns the low-sensitivity function in both cases.

## 3.1 EXPERIMENTS ON SYNTHETIC DATA

We construct a synthetic dataset to examine the inductive bias of a single-layer self-attention model. We show that in the presence of two solutions with the same predictive power but different sensitivity values, this model learns the low-sensitivity function. We begin by describing the experimental setup and then discuss our results.

**Setup.** We compose a single-head self-attention layer with a linear head $\boldsymbol{U} \in \mathbb{R}^{\tilde{d} \times \tilde{d}}$ to obtain the final prediction, and write the full model as

$$\Phi(\boldsymbol{\theta}; \boldsymbol{X}) := \left\langle \boldsymbol{U}, \boldsymbol{\varphi}(\boldsymbol{X}\boldsymbol{W}_Q\boldsymbol{W}_K^{\top}\boldsymbol{X}^{\top})\boldsymbol{X}\boldsymbol{W}_V \right\rangle, \tag{3}$$

where $\boldsymbol{\theta} := \operatorname{concat}(\boldsymbol{W}_Q, \boldsymbol{W}_K, \boldsymbol{W}_V, \boldsymbol{U})$. We consider this model for the experiments in this section, with all the parameters initialized randomly at a small scale. Next, we describe the process to generate the dataset. We first define the vocabulary as follows:

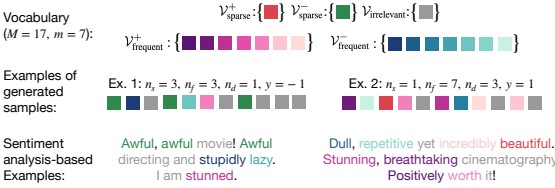

Figure 2: Visualization of the synthetic data generation process (see Section 3.1 for details). For simplicity, we represent each $d$-dimensional token with a square. Middle row: In each case, given a label $y$, we randomly sample $T = 11$ tokens, with $n_s$ tokens from $\mathcal{V}_{\text{sparse}}^y$, $\lfloor (n_f + n_d)/2 \rfloor$ tokens from $\mathcal{V}_{\text{frequent}}^y$, $n_f - \lfloor (n_f + n_d)/2 \rfloor$ tokens from $\mathcal{V}_{\text{frequent}}^{-y}$ and the remaining tokens from $\mathcal{V}_{\text{irrelevant}}$. Note that in the first example, since $n_s = 3$ and $n_d = 1$, a predictor that relies (only) on the sparse tokens is less sensitive compared to the one that relies on the frequent tokens. On the other hand, in the second example, since $n_s = 1$ and $n_d = 3$, the predictor that relies on the frequent tokens is less sensitive. Bottom row: We include two sentiment analysis-based examples to illustrate the synthetic data samples in the second row, using the same colors as the first two rows.

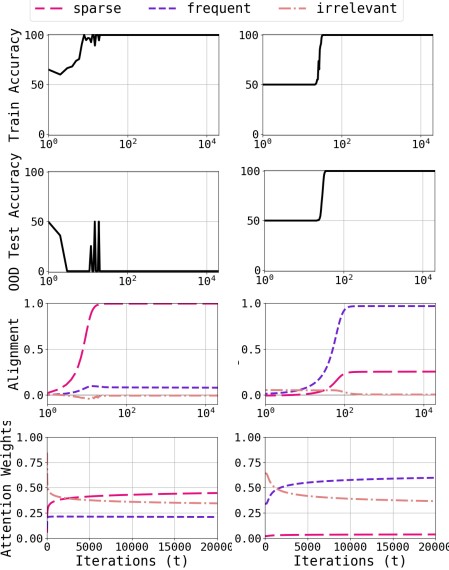

Figure 3: Train and test dynamics for a single-layer self-attention model (Eq. (3)) using the synthetic data visualized in Fig. 2; see Section 3.1 for details. **Left column**: the predictor that uses *sparse* tokens has lower sensitivity (Ex. 1 in Figure 2), **Right column**: the predictor that uses *frequent* tokens has lower sensitivity (Ex. 2 in Figure 2); see Appendix A.1 for more examples.

Table 1: Comparison of sensitivity values for models that use only sparse or frequent tokens for the settings considered in Figure 3.

| $(n_s, n_f, n_d, m)$ | Using sparse tokens | Using frequent tokens |
|---|---|---|
| $(3, 5, 1, 16)$; Fig. 3 left col. | 0 | 0.2878 |
| $(1, 17, 7, 20)$; Fig. 3 right col. | 0.0339 | 0 |

**Definition 3.2** (Synthetic Vocabulary). Consider a vocabulary of $M$ distinct tokens $\mathcal{V} := \{\boldsymbol{e}_1, \dots \boldsymbol{e}_M\}$, where $\boldsymbol{e}_i \in \{0, 1\}^d$ denotes the $i^{\text{th}}$ basis vector. We define smaller subsets of *sparse* tokens and larger subsets of *frequent* tokens for each label $y = \pm 1$, as well as a subset of *irrelevant* tokens:

$$\mathcal{V}_{\text{sparse}}^+ := \{\boldsymbol{e}_1\}, \mathcal{V}_{\text{sparse}}^- := \{\boldsymbol{e}_2\}, \mathcal{V}_{\text{irrelevant}} := \{\boldsymbol{e}_{2m+3}, \dots, \boldsymbol{e}_M\}$$

$$\mathcal{V}_{\text{frequent}}^+ := \{\boldsymbol{e}_3, \boldsymbol{e}_5, \dots, \boldsymbol{e}_{2m+1}\}, \mathcal{V}_{\text{frequent}}^- := \{\boldsymbol{e}_4, \boldsymbol{e}_6, \dots, \boldsymbol{e}_{2m+2}\}.$$

Let $T$ denote the sequence length of each data point, $n_f$ and $n_s$ denote the number of frequent and sparse tokens, respectively, such that $n_s < n_f < \min(m, T - n_s)$, and $n_d$ be a parameter satisfying $n_d \leq n_f$. Next, we describe the process of generating the dataset $\mathcal{D}$; see Fig. 2 for an example.

**Definition 3.3** (Dataset Generation). Consider the vocabulary in Definition 3.2. To generate a data point $(\boldsymbol{X}, y)$, we first sample the label $y \in \{\pm 1\}$ uniformly at random. We divide the indices $[T]$ into three sets $\mathcal{I}_{\text{frequent}}, \mathcal{I}_{\text{sparse}}$ and $\mathcal{I}_{\text{irrelevant}}$, and sample each set as follows:

- $\mathcal{I}_{\text{frequent}}$ is composed of $\lfloor (n_f + n_d)/2 \rceil$ tokens uniformly sampled from $\mathcal{V}_{\text{frequent}}^y$ and $n_f - \lfloor (n_f + n_d)/2 \rceil$ tokens uniformly sampled from $\mathcal{V}_{\text{frequent}}^{-y}$.

- $\mathcal{I}_{\text{sparse}}$ contains $n_s$ tokens uniformly sampled from $\mathcal{V}_{\text{sparse}}^y$.

- The remaining $T - n_f - n_s$ tokens in $\mathcal{I}_{\text{irrelevant}}$ are uniformly sampled from $\mathcal{V}_{\text{irrelevant}}$.

To determine if the tokens in $\mathcal{V}_{\text{sparse}}$ or those in $\mathcal{V}_{\text{frequent}}$ have a more significant impact on the model's predictions, we adapt the test set generation process by altering the second step in Definition 3.3: we sample the sparse tokens from $\mathcal{V}_{\text{sparse}}^{-y}$ instead of $\mathcal{V}_{\text{sparse}}^y$. If this modification leads to a noticeable drop in the test accuracy, it suggests that the model relies on the sparse feature(s) for its predictions.

We consider two other metrics to examine the role of the attention head and the linear predictor. Define three vectors: $\boldsymbol{v}_{\text{sp}} := \boldsymbol{e}_1 - \boldsymbol{e}_2$, $\boldsymbol{v}_{\text{freq}} := \sum_{i \in \mathcal{V}_{\text{frequent}}^+} \boldsymbol{e}_i - \sum_{i \in \mathcal{V}_{\text{frequent}}^-} \boldsymbol{e}_i$, $\boldsymbol{v}_{\text{irrel}} := \sum_{i \in \mathcal{V}_{\text{irrelevant}}} \boldsymbol{e}_i$. We plot the average alignment (cosine similarity) between the rows of $\boldsymbol{U}\boldsymbol{W}_V^\top$ and these vectors to see what tokens the prediction head relies on. Similarly, we plot the sum of the softmax scores for the three types of tokens to see which tokens are selected by the attention mechanism.

We set $\mathcal{P}$ in Def. 3.1 as the uniform distribution over $\mathcal{V}$ for computing sensitivity in our experiments.

**Results.** Figure 3 shows the train and test dynamics of the model in Eq. (3) using synthetic datasets generated by following the process in Definition 3.3 (details in Table 1). We consider two cases: in the first case (left column), using the sparse token leads to a function with lower sensitivity, whereas in the second case (right column), using the frequent tokens leads to lower sensitivity (see Table 1 for a comparison of the sensitivity values). We observe that in the first case, the OOD test accuracy drops to 0, the alignment with $\boldsymbol{v}_{\text{sp}}$ is close to 1 and the attention weights on the sparse tokens are the highest. These results show that the model relies on the sparse token in this case. On the other hand, in the second case, the test accuracy remains high, the alignment with $\boldsymbol{v}_{\text{freq}}$ is close to 1 and the attention weights on the frequent tokens are the highest, which shows that the model relies on the frequent tokens. These results show that the model exhibits a low-sensitivity bias. Note that in both cases, the model can learn a function that relies on a sparse set of inputs (using the sparse tokens), however, it uses these tokens only when doing so leads to lower sensitivity.

## 4 INVESTIGATIONS ON VISION TASKS

In this section, we test whether our notion of sensitivity captures the inductive bias of transformers on vision tasks. We consider Vision Transformers (ViT, Dosovitskiy et al., 2021) which regard images as a sequence of patches instead of a tensor of pixels.

**Definition 4.1** (Tokenization for Vision Transformers). Let $\boldsymbol{X} \in \mathbb{R}^{n_h \times n_w \times n_c}$ be the image with height $n_h$, width $n_w$, and number of channels $n_c$. A tokenization of $\boldsymbol{X}$ is a sequence of $T$ image patches $\{\boldsymbol{e}_1, \cdots, \boldsymbol{e}_T\}$ where each token $\boldsymbol{e}_i$ represents an image patch of dimension $d = n_w n_h n_c / T$.

Sensitivity is measured on the *training* set because our goal is to understand the simplicity bias of the model at training time, to see if it prefers to learn certain simple classes of functions on the training data. Since different models could have different generalization capabilities, the sensitivity on test data might not reflect the model's preference for low-sensitivity functions at training time. Further, since the choice of optimization algorithm could in principle introduce its own bias and our goal is to understand the bias of the architecture, we train both the models with the same optimization algorithm, namely SGD; see Fig. 11 in the App. for a comparison with Adam.

We consider three datasets in this section (see Appendix D for details), namely CIFAR-10 (Krizhevsky, 2009), ImageNet-1k (Russakovsky et al., 2015) and Fashion-MNIST (Xiao et al., 2017). We use $\sigma^2 = 1, 15$ and $5$, respectively. We use a variant of the ViT architecture for small-scale datasets proposed in Lee et al. (2021), referred to as ViT-small here onwards; see Appendix D for more details and Appendix A.3 for additional results where we show that varying model depths and number of heads does not affect the sensitivity of ViT models. We also compare the sensitivity of the ViT-small model and a ResNet-18 (He et al., 2016) CNN on the SVHN dataset with $\sigma^2 = 1$ in Appendix A.3, which leads to the same conclusion.

**Transformers learn lower sensitivity functions than CNNs.** Figure 4 shows the train accuracies as well as the sensitivity comparison between two ViTs: the ViT-small model and a ViT-simple model (Beyer et al., 2022), two CNNs: a ResNet-18 and a DenseNet-121 (Huang et al., 2016), and a ConvMixer model (Trockman & Kolter, 2022a). Note that the train accuracies are comparable for all architectures, which allows for a fair comparison of sensitivity. We observe that the ViTs have significantly lower sensitivity compared to the CNNs and the

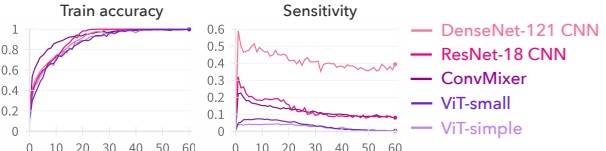

Figure 4: **Sensitivity on CIFAR-10.** Comparison of the sensitivity of two CNNs, two ViTs, and ConvMixer trained on the CIFAR-10 dataset, as a function of training epochs. For a fair comparison, the figure also shows the train accuracies (see App. Fig. 13 for full train dynamics). All models have similar accuracies but the ViTs have significantly lower sensitivity.

ConvMixer model. At the end of the training, the sensitivity values are 0.3673 for DenseNet-121, 0.0827 for ResNet-18, 0.0829 for ConvMixer, 0.0050 for ViT-small and 0.0014 for ViT-simple.

**Do Transformers have lower sensitivity than CNNs because these models process inputs differently?** ViTs process inputs as a sequence of patches whereas CNNs do not, and hence a natural question to ask is if the difference in sensitivity between the two architectures is due to this difference in processing the inputs as opposed to differences in the architecture. To investigate this, we compare ViTs with ConvMixer (Trockman & Kolter, 2022b). Similar to ViTs, ConvMixer processes the input data in a patch-wise manner, but has two key differences: it does not use the self-attention mechanism, which is the core component of transformers, and it relies on convolutions for the feedforward part as well. The higher sensitivity of the ConvMixer model indicates that the low sensitivity simplicity bias of the transformers is not because they process inputs patch-wise, but rather a result of other components of the architecture.

**Do these observations generalize to pre-trained models?** To study this, we consider the ImageNet-1k dataset (Russakovsky et al., 2015). We compare the sensitivity values of pre-trained ConvNext (Liu et al., 2022) and ViT/L-16 (Dosovitskiy et al., 2021) models. For comparable accuracies, ViT/L-16 has a sensitivity of 0.0191, which is lower than that of ConvNext at 0.0342. This shows that the observations on small-scale models studied in this section transfer to large-scale pretrained models.

**Transformers learn lower sensitivity functions than MLPs.** Next, we consider the Fashion-MNIST dataset and compare the sensitivity of ViT-small, a 3-hidden-layer CNN, an MLP with LeakyReLU activation and an MLP with sigmoid activation (see Fig. 15 in the Appendix for the training curves). At the end of training, the sensitivity values are 0.0559 for the MLP with LeakyReLU, 0.0505 for the MLP with sigmoid, 0.0453 for the CNN and 0.0098 for the ViT.

Thus, transformers learn lower sensitivity functions compared to MLPs, ConvMixers, and CNNs.

## 5 INVESTIGATIONS ON LANGUAGE TASKS

In this section, we investigate the sensitivity of transformers on natural language tasks, where each datapoint is a sequence of tokens. Similar to the comparison of ViTs with MLPs and CNNs in Section 4, we compare a RoBERTa (Liu et al., 2019) transformer model with LSTMs (Hochreiter & Schmidhuber, 1997), an alternative auto-regressive model, in this section. Recall that we consider a transformer with linear attention for the results in Section 2. Aligning with this setup, we also consider a RoBERTa model with ReLU activation in the attention layer (*i.e.*, replacing $\varphi(\cdot)$ in Eq. (3) with $\text{ReLU}(\cdot)$) for our experiments.

We use the usual RoBERTa-like tokenization procedure to process inputs for all the models so that they are represented as  $e_1, \cdots, e_T$  where each $e_j$ represents tokens that are usually subwords and  represents the classification (CLS) token, $T$ the sequence length, and  the separator token. We denote $e_0 =$  and $e_{T+1} =$ . For each token $e_j$, a token embedding $h_E(\cdot) : [M] \to \mathbb{R}^d$ is trained during the process, where $M$ denotes the vocabulary size. For transformers, we also train a separate positional encoder $h_P(\cdot) : [N] \to \mathbb{R}^d$, where $N$ denotes the maximum sequence length. We denote $e_j^{\text{LSTM}} = h_E^{\text{LSTM}}(e_j)$ and $e_j^{\text{RoBERTa}} = h_E^{\text{RoBERTa}}(e_j) + h_P^{\text{RoBERTa}}(j)$ as the embedding tokens of LSTM and RoBERTa, respectively. We omit the superscript for convenience.

To control the relative magnitude of noise, the embeddings $e_\tau \leftarrow \text{LayerNorm}(e_\tau)$ are first layer-normalized (Ba et al., 2016) before the additive Gaussian corruption. To better control possible

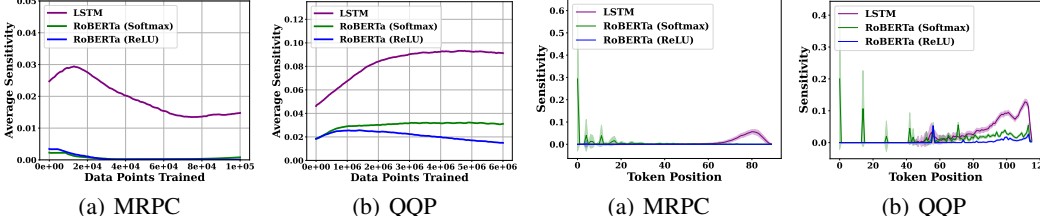

Figure 5: **Sensitivity over Datapoints Trained**. On both datasets, the Transformer-based model RoBERTa displays much lower sensitivity compared to LSTMs during the entire training process. RoBERTa with ReLU activation has lower sensitivity compared to its Softmax counterpart at later stages of training.

Figure 6: **Sensitivity over Token Position**. On both datasets, LSTMs are more sensitive to later tokens than early ones, while RoBERTa's sensitivity, regardless of the activation function, is more uniform across token positions, except for a few early bumps in early tokens which come from the CLS token .

confounders, we limit both LSTM and RoBERTa to having the same number of layers. Both models are trained from scratch, *without* any pretraining on larger corpora, to ensure fair comparisons.

We consider two binary classification datasets, MRPC (Dolan & Brockett, 2005) and QQP (Iyer et al., 2017) (see Appendix D for details), which are relatively easy to learn without pretraining (Kovaleva et al., 2019). Empirically, we set $\sigma^2 = 15$ (results with $\sigma^2 = 4$ in App. A.4 yield similar observations as the results in this section). Similar to Section 4, we measure sensitivity on the train set (results on the validation set in App. A.4 yield similar observations). We include results with different depth values for RoBERTa as well as using GPT-2 in App. A.4 and they lead to similar conclusions.

**Transformers learn lower sensitivity functions than LSTMs.** As shown in Figure 5, both RoBERTa models have lower sensitivity than LSTMs on both datasets, regardless of the number of datapoints trained. Even at initialization with random weights, LSTMs are more sensitive. At the end of training, the sensitivity values on the MRPC dataset are 0.15, 0.002 and 0.001 for the LSTM, the RoBERTa model with softmax activation and the RoBERTa with ReLU activation, respectively. On the QQP dataset, LSTM, RoBERTa-softmax and RoBERTa-ReLU have sensitivity values of 0.09, 0.03 and 0.02, respectively. Interestingly, RoBERTa with ReLU activation also has lower sensitivity than its softmax counterpart. This may be because softmax attention encourages sparsity because of which the model can be more sensitive to a particular token; see Ex. 2 in Fig. 2 and bottom row of Fig. 3 for an example where sparsity can lead to higher sensitivity.

**LSTMs are more sensitive to later tokens.** In Fig. 6, we plot sensitivity over the token positions. We observe that LSTMs exhibit larger sensitivity towards the end of the sequence, i.e. at later token positions. In contrast, transformers are relatively uniform. Similar observations were made by (Fu et al., 2023) for a linear regression setting: LSTMs do more local updates and only remember the most recent observations, whereas transformers preserve global information and have longer memory.

**Transformers are sensitive to the CLS token.** In Fig. 6, we also observe that the RoBERTa model with softmax activation has frequent bumps in the sensitivity values at early token positions. This is because different sequences have different lengths and while computing sensitivity versus token positions, we align all the sequences to the right. These bumps at early token positions indeed correspond to the starting token after the tokenization procedure, the CLS token . This aligns with the observation of Jawahar et al. (2019) that the CLS token gathers all global information. Perturbing the CLS token corrupts the aggregation and results in high sensitivity. We also observe that RoBERTa with ReLU activation seems less sensitive to the CLS token compared to its softmax counterpart.

## 6 IMPLICATIONS OF LOW SENSITIVITY BIAS

We saw in Section 4 that transformers learn lower sensitivity functions than CNNs. In this section, we first compare the test performance of these models on the CIFAR-10-C dataset and show that transformers are more robust than CNNs. Next, we add a regularization term while training the transformer, to encourage lower sensitivity. The results demonstrate that lower sensitivity leads to improved robustness. We then explore the connection between sensitivity and the flatness of the minima. Our results show that lower sensitivity leads to flatter minima. Finally, we examine if

sensitivity can be used to understand the training dynamics of transformers, where we find sensitivity to be a suitable progress measure for certain grokking instances.

## 6.1 LOWER SENSITIVITY LEADS TO IMPROVED ROBUSTNESS

The CIFAR-10-C dataset (Hendrycks & Dietterich, 2019) was developed to benchmark the performance of various NNs on object recognition tasks under common corruptions that are not confusing to humans. Images from the test set of CIFAR-10 are corrupted with 14 types of algorithmically generated corruptions from blur, noise, weather, and digital categories (see Fig. 1 in Hendrycks & Dietterich (2019) for examples).

Fig. 7 compares the performance of two CNNs: ResNet-18 and DenseNet-121 with two ViTs: ViT-small and ViT-simple on various corruptions from the CIFAR-10-C dataset, at the end of training. We observe that the ViTs have lower sensitivity and better test performance on almost all corruptions compared to the CNNs, which have a higher sensitivity. Since the definition of sensitivity involves the addition of noise and ViTs have lower sensitivity, one can expect to be robust to various noise corruptions. However, the ViTs also have better test performance on several corruptions from weather and digital categories, which are significantly different from noise corruptions. This is consistent with the observations in Mahmood et al. (2021); Bhojanapalli et al. (2021).

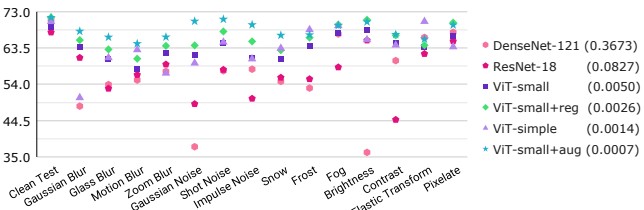

Figure 7: Comparison of the test accuracies on CIFAR-10 and on various corruptions from the CIFAR-10-C dataset (see Section 6 for details) of various models trained on the CIFAR-10 dataset, at the last training epoch (see App. Fig. 17 for a comparison of the accuracies as a function of training epochs.). We observe that the Vision Transformer models `ViT-small` and `ViT-simple` exhibit lower sensitivity and higher robustness to corruptions compared to the CNN models `DenseNet-121` and `ResNet-18`. Additionally, encouraging lower sensitivity while training through regularization (`ViT-small-reg`) and data augmentation (`ViT-small-aug`) leads to improved robustness (see Section 6 for details).

Next, we conduct an experiment to investigate the role of low sensitivity in the robustness of transformers. We add a regularization term while training the model to explicitly encourage it to have lower sensitivity. If this model is more robust, then we can disentangle the role of low sensitivity from the role of the architecture and establish a concrete connection between lower sensitivity and improved robustness. To add the regularization, we use the fact that sensitivity can be estimated efficiently via sampling and consider two methods. In the first method (augmentation), we augment the training set by injecting the images with Gaussian noise (mean $0$, variance $0.1$) while preserving the label, and train the ViT on the augmented training set. In the second method (regularization), we add a mean squared error term using the model outputs for the original image and the image with Gaussian noise (mean $0$, variance $1$) injected into a randomly selected patch.

Fig. 7 also shows the test performance of ViT-small trained with augmentation and regularization methods on various corruptions from CIFAR-10-C. We observe that ViTs trained with these methods exhibit lower sensitivity compared to vanilla training. This is accompanied by an improved test performance on various corruptions, particularly on the noise and blur categories. As encouraging lower sensitivity improves robustness, the inductive bias of transformers to learn functions of lower sensitivity could explain their better robustness (to common corruptions) compared to CNNs.

## 6.2 LOWER SENSITIVITY LEADS TO FLATTER MINIMA

In this section, we investigate the connection between low sensitivity and flat minima. Consider a linear model $\Phi(\boldsymbol{\theta}; \boldsymbol{x}) = \boldsymbol{\theta}^\top \boldsymbol{x}$. Measuring sensitivity involves perturbing the input by some $\Delta \boldsymbol{x}$. Prediction on the perturbed input is equivalent to perturbing the weight vector with $\Delta \boldsymbol{\theta} = \frac{\boldsymbol{\theta}^\top \Delta \boldsymbol{x}}{\|\boldsymbol{x}\|_2^2} \boldsymbol{x}$, as

$$\Phi(\boldsymbol{\theta}; \boldsymbol{x} + \Delta \boldsymbol{x}) = \boldsymbol{\theta}^\top (\boldsymbol{x} + \Delta \boldsymbol{x}) = \Phi(\boldsymbol{\theta}; \boldsymbol{x}) + \boldsymbol{\theta}^\top \Delta \boldsymbol{x} = \Phi(\boldsymbol{\theta}; \boldsymbol{x}) + \Delta \boldsymbol{\theta}^\top \boldsymbol{x} = \Phi(\boldsymbol{\theta} + \Delta \boldsymbol{\theta}; \boldsymbol{x}).$$

This draws a natural connection between sensitivity, which is measured with perturbation in the input space, and flatness of minima, which is measured with perturbation in the weight space (Keskar et al.,

2017). Below, we investigate whether such a connection extends to more complex architectures such as transformers. Given model $\Phi$ and train set $\mathcal{D}$, we consider two metrics to measure the flatness of the minimum, based on the model outputs and model predictions, respectively,

$$\text{ShOp} := \mathop{\mathbb{E}}_{\boldsymbol{x}\sim\mathcal{D},\boldsymbol{\xi}\sim\mathcal{N}(0,\sigma^2\boldsymbol{I})} |\Phi(\boldsymbol{\theta};\boldsymbol{x}) - \Phi(\boldsymbol{\theta}+\boldsymbol{\xi};\boldsymbol{x})|, \text{ ShPred} := \mathop{\mathbb{E}}_{\boldsymbol{x}\sim\mathcal{D},\boldsymbol{\xi}\sim\mathcal{N}(0,\sigma^2\boldsymbol{I})} \mathbb{1}[f(\boldsymbol{\theta};\boldsymbol{x}) \neq f(\boldsymbol{\theta}+\boldsymbol{\xi};\boldsymbol{x})],$$

where $f(\boldsymbol{\theta};\boldsymbol{x}) = \mathbb{1}[\Phi(\boldsymbol{\theta};\boldsymbol{x}) \geq 0]$. Intuitively, for flatter minima, the model output and hence its prediction would remain relatively invariant to small perturbations in the model parameters.

Table 2 shows a comparison of these metrics for the ViT-small model trained with and without the sensitivity regularization at the end of training. Both metrics indicate that lower sensitivity corresponds to a flatter minimum. It is widely believed that flatter minima correlate with better generalization (Jiang* et al., 2020; Keskar et al., 2017; Neyshabur et al., 2017), though they may not always be correlated (Andriushchenko et al., 2023). Our results indicate that low-sensitivity

Table 2: Comparison of two sharpness metrics at the end of training the ViT-small model on the CIFAR-10 dataset with and without the sensitivity regularization. Lower values correspond to flatter minima; see text for discussion.

| Setting | ShOp | ShPred |
|---|---|---|
| ViT-small + vanilla training | 39.166 | 0.5346 |
| ViT-small + sensitivity regularization | 9.025 | 0.3982 |

correlates with improved generalization and investigating this connection for other settings can be an interesting direction for future work.

### 6.3 SENSITIVITY AS A PROGRESS MEASURE FOR GROKKING

In this section, we investigate if the sensitivity notion could serve as a progress measure for grokking (Nanda et al., 2023; Chen et al., 2024). We train an one-layer Transformer model on the modular addition task $a + b \mod 113$. When evaluating sensitivity, we add a random Gaussian noise

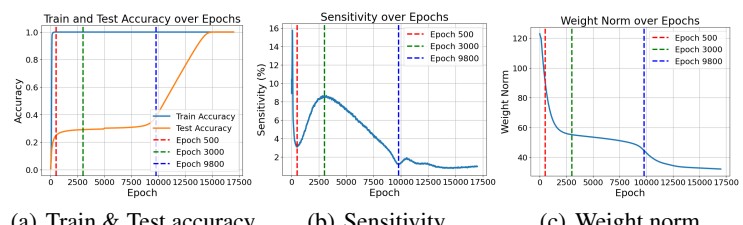

(a) Train & Test accuracy     (b) Sensitivity     (c) Weight norm

Figure 8: Sensitivity measures progress on modular addition task $a + b \mod 113$ and indicates different stages of grokking.

with $\sigma = 0.1$ to the number embeddings. As shown in Figure 8, the test accuracy stays low from epoch 500 to 9,800 while the training accuracy saturates. However, sensitivity values continue to decrease smoothly starting from epoch 3000, and hence it provides a measure of the *hidden progress* (Barak et al., 2022) which the model makes even though the loss does not change, and indicates stages of grokking. In contrast, the weight norm is not a progress measure since it has the same flat curve as test accuracy during epoch 3,000 to 9,800.

As discussed by Nanda et al. (2023), grokking occurs when the model learns to use Fourier features to solve the task. A further bump in sensitivity after the grokked phase at epoch 9800 suggests that the model initially learns less robust Fourier features. At this stage, a small random noise could slightly disrupt the model's performance. Over time, the Fourier basis becomes more robust. See Appendix A.5 for further discussion and results for more settings.

### 7 CONCLUSION

In this work, we investigate how the notion of sensitivity, which has shown promise in understanding inductive biases of Transformers on Boolean functions in prior work, can be extended to more realistic settings involving real-valued data. Our results show that transformers learn functions that have low sensitivity to small token-wise input perturbations, compared to other architectures, across vision and language tasks. We corroborate these observations with theoretical results, showing that transformers exhibit spectral bias and lower sensitivity corresponds to better robustness. We also demonstrate three important implications of this low-sensitivity bias: it correlates with improved robustness, flatter minima in the loss landscape, and serves as a progress measure that offers insights about the training dynamics. Investigating sensitivity as a progress measure in more settings can be an interesting direction for future work.

ACKNOWLEDGEMENTS

The authors thank the anonymous reviewers for helpful suggestions and feedback. This work was supported by an NSF CAREER Award CCF-2239265, an Amazon Research Award, an Open Philanthropy research grant, and a Google Research Scholar Award. YH was supported by a USC CURVE fellowship. The authors acknowledge the use of USC CARC's Discovery cluster and the USC NLP cluster. This work was done in part while BV, DF, TZ, and VS were visiting the Simons Institute for the Theory of Computing.

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

APPENDIX

## A    ADDITIONAL EXPERIMENTS

In this section, we include some additional results to supplement the main experimental results for synthetic data as well as the vision and language tasks.

### A.1    SYNTHETIC DATA AND THE MNIST DATASET

In this section, we present some additional results for the low-sensitivity bias of a single-layer self-attention model (Eq. (3)) on the synthetic dataset generated based on Definition 3.3, visualized in Fig. 2. Similar to the results in Section 3.1, we consider three data settings where using the sparse token leads to a function with lower sensitivity (Fig. 9, top row) and three settings where using the frequent token leads to lower sensitivity (Fig. 9, bottom row). The exact data settings and a comparison of the sensitivity values for each setting are shown in Table 3. These results yield similar conclusions as in Section 3.1: in both cases, the model uses tokens which leads to a lower sensitivity function.

Continuing from the synthetic data, we now consider a slightly more complicated dataset, namely MNIST (LeCun & Cortes, 2005). The MNIST dataset consists of $70k$ black-and-white images of handwritten digits of resolution $28 \times 28$. There are $60k$ images in the training set and $10k$ images in the test set. It is released under the CC BY-SA 3.0 license. We compare the sensitivity of a ViT-small model with an MLP on a binary digit classification task ($< 5$ or $\geq 5$). In our experiments, each image is divided into $T = 16$ patches of size $7 \times 7$ for the ViT-small model. For the MLP, the inputs are vectorized as usual. With this setting, we measure the sensitivity of the two models using patch token replacement as per Definition 3.1. As shown in Figure 10, when achieving the same training accuracy, the ViT shows lower sensitivity compared to the MLP.

### A.2    SENSITIVITY WITH RANDOM NOISE INSTEAD OF TOKEN-WISE NOISE

In this section, we consider changing the way we compute sensitivity, to see if the resulting metric also distinguishes transformers from other architectures. Instead of token-wise perturbations, we add

Table 3: Comparison of sensitivity values for models that use only sparse or frequent tokens for the settings considered in Fig. 9.

| Data Setting $(n_f, m)$ | Top row in Fig. 3 ($n_s = 3, n_d = 1$) | | | Bottom row in Fig. 3 ($n_s = 1, n_d = 7$) | | |
|---|---|---|---|---|---|---|
| | $(3, 6)$ | $(5, 16)$ | $(7, 28)$ | $(7, 10)$ | $(17, 20)$ | $(32, 36)$ |
| Using sparse tokens | 0 | 0 | 0 | 0.0339 | 0.0339 | 0.0339 |
| Using frequent tokens | 0.1315 | 0.2878 | 0.4502 | 0 | 0 | 0 |

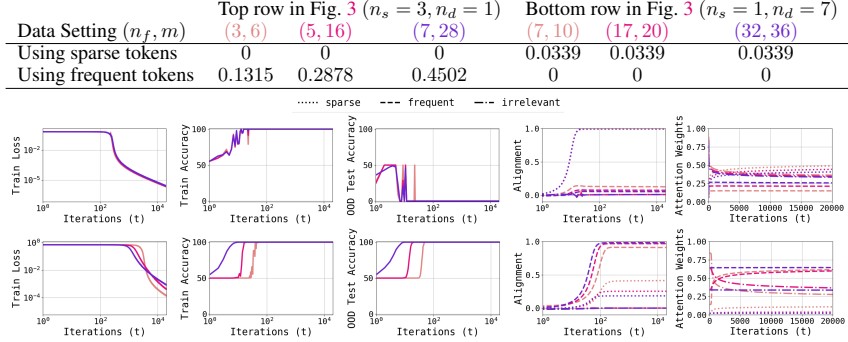

Figure 9: Train and test dynamics for a single-layer self-attention model (Eq. (3)) using the synthetic data visualized in Fig. 2; see Section 3.1 for details. The top row corresponds to the cases where the predictor that uses sparse tokens has lower sensitivity, while the bottom row corresponds to the cases where using the frequent tokens leads to lower sensitivity. The precise data settings for this figure, as well as a comparison of sensitivity values, are shown in Table 3.

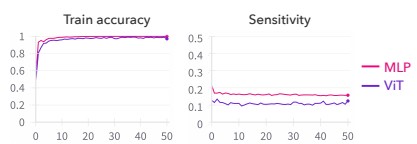

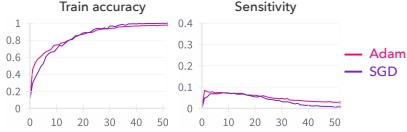

Figure 10: **Sensitivity on MNIST.** ViT and MLP get similar accuracy, but the ViT has lower sensitivity.

Figure 11: **Sensitivity using SGD and Adam.** Comparison of train accuracies and sensitivity values of the ViT-small model trained on the CIFAR-10 dataset using SGD and Adam optimizers.

Gaussian noise across the entire input with a smaller variance so that the transformer's sensitivity in this case is similar to the sensitivity with token-wise noise.

Table 4: Comparison of sensitivity values measured with random and patch-wise noise for various model-dataset settings. Token-wise perturbations lead to a larger gap between the sensitivity of transformer-based models compared to other architectures.

| Model and dataset | Random noise | Token-wise noise |
|---|---|---|
| ResNet-18 on CIFAR-10 | 0.0172 | 0.0827 |
| ViT-small on CIFAR-10 | 0.0082 | 0.0050 |
| LSTM on QQP | 0.11 | 0.09 |
| RoBERTa on QQP | 0.05 | 0.03 |

In Table 4, we compare the sensitivity values at the end of training for ResNet-18 and ViT-small on the CIFAR-10 dataset (variance 0.025) and LSTM and RoBERTa on the QQP dataset (variance 0.5). We find that for random perturbations, the difference between sensitivity values is much smaller for CIFAR-10 and similar for the QQP dataset, compared to patch-wise perturbations. These results suggest that measuring sensitivity with patch-wise noise is indeed the metric that we should consider since it distinguishes transformers from other architectures with a larger gap.

## A.3 VISION TASKS

**Effect of Depth, Number of Heads and the Optimization Algorithm.** In Fig. 11, we compare the sensitivity values of a ViT-small model trained on CIFAR-10 dataset with SGD and Adam

optimization algorithms. Although the model trained with Adam has a slightly higher sensitivity, the sensitivity values for both the models are quite similar. This indicates that the low-sensitivity bias is quite robust to the choice of the optimization algorithm.

In Fig. 12, we compare the sensitivity values of a ViT-small model with different depth and number of attention heads, when trained on the CIFAR-10 dataset. Note that for our main results, we use a model with depth 8 and 32 heads. We observe that the train accuracies and the sensitivity values remain the same across the different model settings. This indicates that the low-sensitivity bias is quite robust to the model setting.

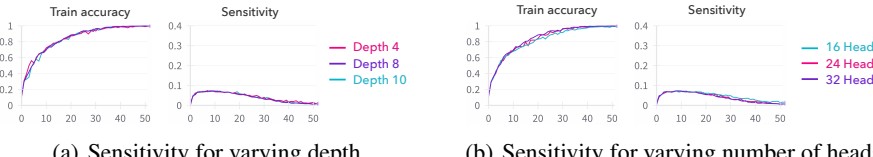

    (a) Sensitivity for varying depth         (b) Sensitivity for varying number of heads

Figure 12: **Sensitivity for Various Model Settings.** Comparison of train accuracies and sensitivity values on the CIFAR-10 dataset when varying the depth and number of heads of the ViT-small model. We observe that for the same train accuracy, the sensitivity values remain very similar for different model settings.

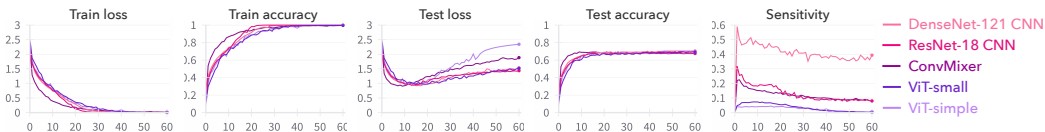

Figure 13: Comparison of the sensitivity of two CNNs, two ViTs, and ConvMixer trained on the CIFAR-10 dataset, as a function of training epochs. For a fair comparison, the figure also shows the train and test accuracies and loss values (cross-entropy loss). All models have similar accuracies but the ViTs have significantly lower sensitivity than the other models.

**Effect of Variance.** In Fig. 14, we compare the effect of the variance $\sigma^2$ used while evaluating sensitivity for different models trained on the CIFAR-10 dataset. We observe that the ViTs have significantly lower sensitivity than the other models and the difference becomes starker as the variance level increases.

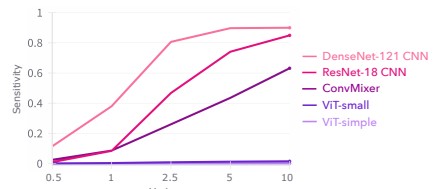

**Results on SVHN Dataset.** Fig. 16 shows the training accuracy and sensitivity of a ResNet-18 and a ViT-small model trained on SVHN dataset (Netzer et al., 2011). At the end of training, the sensitivity values are: 0.0516 for ResNet-18 and 0.0147 for ViT-small. Similar to the observations for CIFAR-10, we see that the ViT has a significantly lower sensitivity.

Figure 14: **Sensitivity for Different Variances.** Comparison of the sensitivity of two CNNs, two ViTs, and ConvMixer trained on the CIFAR-10 dataset, as a function of different variance levels, at the end of training. The ViTs have significantly lower sensitivity at any variance and the difference grows as variance increases.

**Additional Results on CIFAR-10-C.** Fig. 17 and Fig. 18 show the test performance on various corruptions from the CIFAR-10-C dataset with severity level 2 and 1, respectively. We observe that CNNs have lower test accuracies on corrupted images compared to ViTs. Further, encouraging lower sensitivity in the ViT leads to better robustness.

## A.4 LANGUAGE TASKS

**Sensitivity Measured with Variance $\sigma^2 = 4$.** Alternative to the main experiments with $\sigma^2 = 15$, we also evaluate sensitivity with a different corruption strength $\sigma^2 = 4$ on the QQP dataset, as shown in

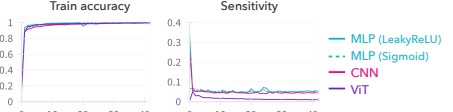 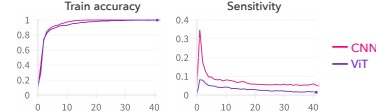

Figure 15: **Sensitivity on Fashion-MNIST.** Comparison of sensitivity of a ViT with a CNN, an MLP with LeakyReLU activation and an MLP with sigmoid activation, as a function of training epochs. All the models have similar accuracies but the ViT has significantly lower sensitivity.

Figure 16: **Sensitivity on SVHN.** Comparison of sensitivity of a ResNet-18 CNN and a ViT-small trained on SVHN dataset, as a function of training epochs. Both the models have similar accuracies but the ViT has significantly lower sensitivity.

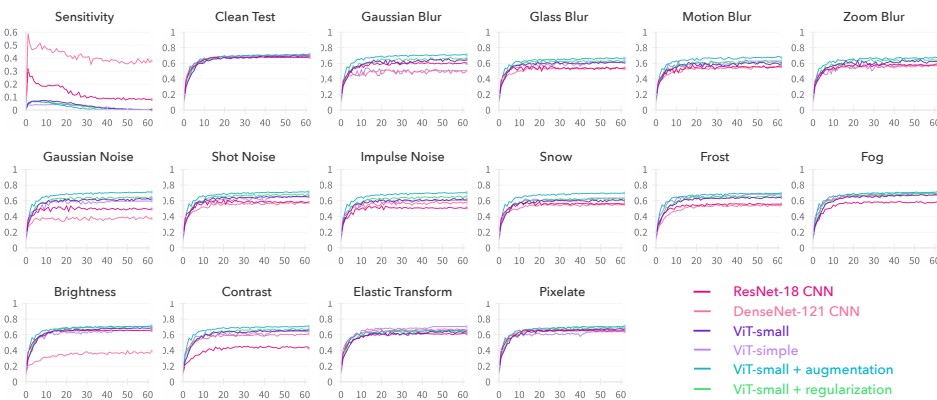

Figure 17: Comparison of the sensitivity, test accuracy on CIFAR-10 and test accuracies on various corruptions from the CIFAR-10-C dataset (see Section 6 for details) of two CNNs and two ViTs trained on the CIFAR-10 dataset, as a function of the training epochs. We also compare with ViT-small trained with data augmentation/regularization, which encourage low sensitivity (see Section 6 for discussion).

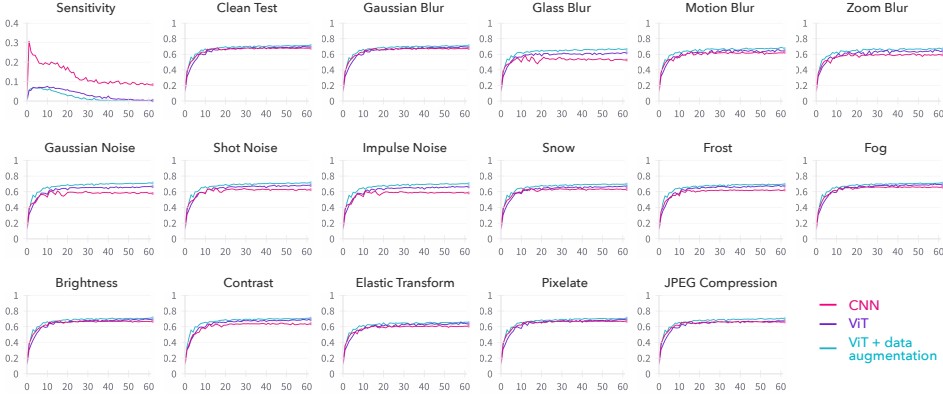

Figure 18: Comparison of the sensitivity, test accuracy on CIFAR-10, and test accuracies on various corruptions from the CIFAR-10-C dataset (see Section 6 for details) of a ResNet-18 CNN and a ViT-small model trained on the CIFAR-10 dataset, as a function of the training epochs. We also compare with ViT-small trained with data augmentation, which acts as a regularizer to encourage low sensitivity (see Section 6 for discussion). Here, we use severity level 1, while in Figure 17, we considered severity level 2.

Fig. 19. We observe the same trends as in Figures 5 and 6: RoBERTa models have lower sensitivity than the LSTM and the LSTM is more sensitive to more recent tokens. These results indicate that the low-sensitivity bias is robust to the choice of corruption strength.

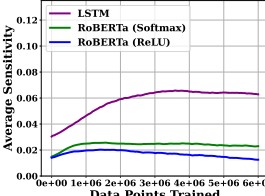 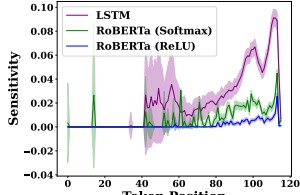

Figure 19: **Sensitivity on the QQP Dataset with Variance** $\sigma^2 = 4$**.** Results with alternative variance yield observations that are consistent with the setup in the main text. (Left) LSTM has higher sensitivity than the RoBERTa models. (Right) Softmax activation for RoBERTa induces higher sensitivity towards the CLS token.

**Sensitivity Measured on the Validation Set.** We also test sensitivity on the validation set for the two language datasets. As seen in Figure 20, the results on the validation set are consistent with those on the train set in Fig. 5.

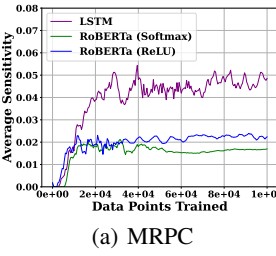 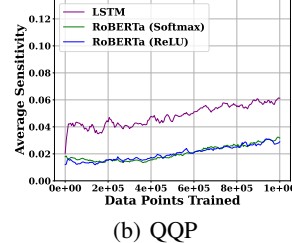

(a) MRPC        (b) QQP

Figure 20: **Sensitivity on the Validation Sets.** Similar to the observation in Figure 5, the RoBERTa models have lower sensitivity than the LSTM for both the datasets. However, the difference between RoBERTa-ReLU and RoBERTa-softmax is less marginal on the validation set compared to the training set.

**Sensitivity Measured with the GPT-2 Model.** Here, we ablate the effect of the transformer architecture on the sensitivity values. We compare a GPT-based model with the two BERT-based models used in our main experiments. The key difference lies in the construction of the attention masks: for GPT models, each token only observes the tokens that appear before it, whereas BERT models are bidirectional, therefore each token observes all the tokens in the sequence. In Fig. 21, we observe that the GPT-2 model has higher sensitivity compared to the RoBERTa models, but the sensitivity is significantly lower than the LSTM. The GPT-2 model is also relatively more sensitive to more recent tokens compared to the RoBERTa models, while also being sensitive to some CLS tokens.

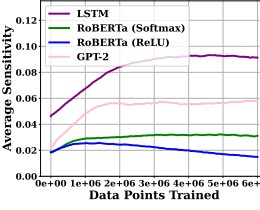 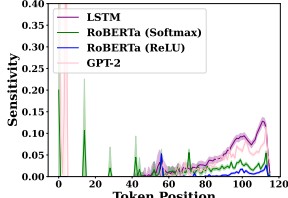

Figure 21: **Sensitivity of GPT-2 on the QQP Dataset.** (Left) We find that the RoBERTa models tend to have lower sensitivity than GPT-2, and all Transformer models have lower sensitivity than LSTM. (Right) The sensitivity per token of GPT-2 is more similar to LSTMs, which is possibly due to their shared auto-regressive design.

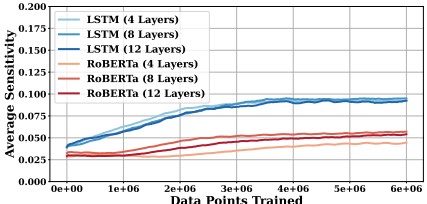

Figure 22: **Sensitivity for Different Model Depths**. We vary the model depths of LSTM and RoBERTa on the QQP datasets and observe that LSTM models tend to have the same sensitivity throughout the entire training. RoBERTa model with 4 layers has slightly lower sensitivity with its 8-layer or 12-layer variants. RoBERTa, regardless of depths, have lower sensitivity than LSTMs.

## A.5 SENSITIVITY AS A PROGRESS MEASURE FOR GROKKING

Grokking is a phenomenon in which a neural network suddenly and drastically improves its generalization ability after a long period of training, even though it initially overfits to the training data (Nanda et al., 2023). During grokking, the model transitions from memorizing the training data to learning a more general solution, allowing it to perform well on unseen data. This often happens after many training steps, during which time the test accuracy remains low despite perfect training accuracy. The transition is abrupt, making grokking seem like an emergent behavior where the model, after much training, "figures out" the correct approach to the task.

In more technical terms, this shift occurs as the network amplifies structured mechanisms that enable generalization and removes components that only lead to memorization. Grokking has been observed in models trained with regularization techniques like weight decay on algorithmic tasks, where the model learns an underlying structure that generalizes well beyond the training data. In (Nanda et al., 2023), the authors demonstrate that for the modular addition task, the model initially memorizes the training data, leading to low test accuracy. Later, the model discovers how to use trigonometric functions to solve the task. However, the emergence of grokking is difficult to measure in practice, and it is not guaranteed that grokking will occur or that the model will find the correct approach to the task. Nevertheless, as shown in Figure 23, we can clearly observe a significant change in sensitivity between epochs 3000 and 9800, during the saturation of the training loss. This suggests that the model is discovering a more robust solution to the task. We propose that sensitivity can serve as a useful metric for assessing whether the model is learning to grok.

Similar to Nanda et al. (2023)'s classification on phases: memorization, circuit formation, and cleanup, we claim that sensitivity also provides a progress measure, with an extra phase of noise reduction:

**Memorization:** From epoch 0 to 500, sensitivity drops significantly while training accuracy saturates to 100 and test accuracy increases but cannot saturate.

**Circuit Formation:** From epoch 500 to 3,000, the sensitivity goes up again but test accuracy remains low and flat. The dramatic fall in the weight norms suggests that circuit formation likely happens due to weight decay.

**Clean up:** From epoch 3,000 to 9,800, sensitivity starts to decrease, and the progress measure indicates that the model starts to learn to use Fourier features.

**Noise Reduction:** From epoch 9,800 onwards, sensitivity has an upward and then downward trend, and this is not caught by other mechanistic interpretability measures in Nanda et al. (2023). This trend in sensitivity is due to further Fourier noise reduction, where initially, slightly perturbations to the number embeddings could drift the model performance where later on, model learns a more robust Fourier basis.

## A.6 SENSITIVITY AS A PROGRESS MEASURE FOR LEARNING SPARSE PARITIES

In this section, we investigate if sensitivity also acts as a progress measure when training transformers on the sparse parity task, with input $x \in \{\pm 1\}^d$ and label $y = \prod_{i \in S} x_i$, where $S \subset [d]$ with sparsity level $p := |S| < d$.

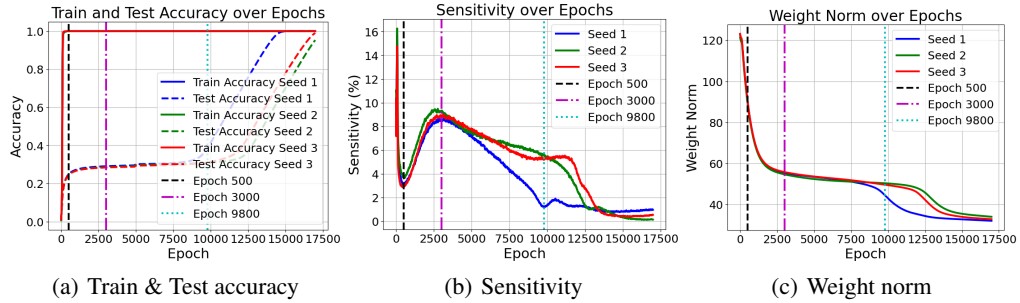

(a) Train & Test accuracy     (b) Sensitivity     (c) Weight norm

Figure 23: Sensitivity measures progress on modulo addition task $a + b \mod 113$ and indicates different stages of grokking.

We train a two-layer transformer model on this task with 4 heads using Adam optimizer for 1000 epochs. We evaluate the sensitivity metric based on Eq. (1), using $10^5$ samples to estimate the expectation over the Boolean cube. Fig. 24 shows the results on six different settings in terms of (number of train samples, batch size, embedding dimension, learning rate, seed):

- Sea green: $(50k, 250, 32, 0.001, 123)$    - Parrot green: $(50k, 250, 32, 0.001, 42)$
- Dark green: $(50k, 250, 32, 0.001, 0)$    - Maroon: $(10k, 100, 64, 0.001, 42)$
- Coral: $(5k, 25, 32, 0.0001, 42)$    - Pink: $(5k, 50, 32, 0.0001, 42)$.

We observe that in all the settings, the train and test accuracy remain close to $50\%$ for the initial few epochs, while sensitivity increases, which indicates progress in learning the sparse parity task. When the model learns the sparse parity function, the sensitivity converges to $\frac{p}{d} = 0.1$, which coincides with train and test accuracy going to $100\%$.

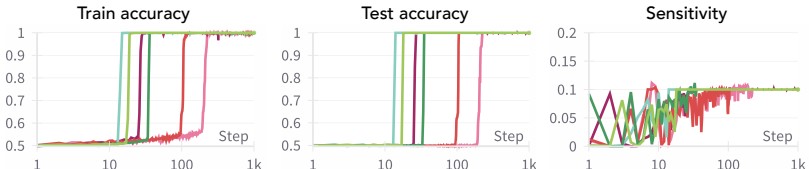

Figure 24: Train accuracy, test accuracy, and sensitivity as a function of training epochs, when training a two-layer transformer on the sparse parity task with dimension 40 and sparsity level 4. Sensitivity increases while train and test accuracy are close to $50\%$, and converges to 0.1 when the model learns the sparse parity function, which coincides with train and test accuracy going to $100\%$.

# B PROOFS FOR SECTION 2

We give a brief overview of the CK and NTK here and refer the reader to (Lee et al., 2018; Yang & Salman, 2020) for more details.

Consider a model with $L$ layers and widths $\{d_l\}_{l=1}^L$ and an input $\boldsymbol{x}$. Let $g^l(\boldsymbol{x})$ denote the output of the $l^{\text{th}}$ layer scaled by $d_l^{-1/2}$. Suppose we randomly initialize weights from the Gaussian distribution $\mathcal{N}(0, 1)$. It can be shown that in the infinite width limit when $\min_{l \in [L]} d^l \to \infty$, each element of $g^l(\boldsymbol{x})$ is a Gaussian process (GP) with zero mean and kernel function $K^l$. The kernel $K^L$ corresponding to the last layer of the model is the CK. In other words, it is the kernel induced by the embedding $\boldsymbol{x} \mapsto g^{L-1}(\boldsymbol{x})$ when the model is initialized randomly. On the other hand, NTK corresponds to training the entire model instead of just the last layer. Intuitively, when the model parameters $\boldsymbol{\theta}$ stay close to initialization $\boldsymbol{\theta}_0$, the residual $g^L(\boldsymbol{x}; \boldsymbol{\theta}) - g^L(\boldsymbol{x}; \boldsymbol{\theta}_0)$ behaves like a linear model with features given by the gradient at random initialization, $\nabla_{\boldsymbol{\theta}} g^L(\boldsymbol{x}, \boldsymbol{\theta}_0)$, and the NTK is the kernel of this linear model. The spectra of these kernels provide insights about the implicit prior of a randomly initialized

model as well as the implicit bias of training using gradient descent (Yang & Salman, 2020). The closer these spectra are to the spectrum of the target function, the better we can expect training using gradient descent to generalize.

## B.1 PROOF OF PROPOSITION 2.1

(Hron et al., 2020) show that the self-attention layer with linear attention and $d^{-1/2}$ scaling converges in distribution to $\mathcal{GP}(0, K)$ in the infinite width limit, *i.e.* when the number of heads $d^H$ become large. For any layer $l \in [L]$, let $\tilde{K}^l$ denote the kernel induced by the intermediate transformation when applying some nonlinearity $\phi$ to the output of the previous layer $l-1$. Let $f^l_{\cdot j} := \{f^l_{i,j}(\boldsymbol{x}) : \boldsymbol{x} \in \mathcal{X}, i \in [T]\}$, where $\mathcal{X}$ denotes the input space of $\boldsymbol{x}$. They show the following result for NNs with at least one linear attention layer, in the infinite width limit.

**Theorem B.1** (Theorem 3 in (Hron et al., 2020)). *Let $l \in [L]$, and $\phi$ be such that $|\phi(x)| \leq c + m|x|$ for some $c, m \in \mathbb{R}^+$. Assume $g^{l-1}$ converges in distribution to $g^{l-1} \sim GP(0, K^{l-1})$, such that $g^{l-1}_{\cdot j}$ and $g^{l-1}_{\cdot k}$ are independent for any $j \neq k$. Then as $\min\{d^{l,H}, d^l\} \to \infty$, $g^l$ converges in distribution to $g^l \sim GP(0, K^l)$ with $g^l_{\cdot k}$ and $g^l_{\cdot \ell}$ independent for any $k \neq \ell$, and*

$$K^l(\boldsymbol{x}, \boldsymbol{x}') = \mathbb{E}[g^l(\boldsymbol{x}) g^l(\boldsymbol{x}')] = \sum_{i,j=1}^{\tilde{d}} (\tilde{K}^l_{ij}(\boldsymbol{x}, \boldsymbol{x}'))^2 \tilde{K}^l_{ab}(\boldsymbol{x}, \boldsymbol{x}').$$

Similar results are also known for several non-linearities and other layers such as convolutional, dense, average pooling (Lee et al., 2018; de G. Matthews et al., 2018; Garriga-Alonso et al., 2019; Novak et al., 2019; Yang, 2021), as well as residual, positional encoding and layer normalization (Hron et al., 2020; Yang, 2021).

Consequently, any model composed of these layers, such as a transformer with linear attention, also converges to a Gaussian process. This follows using an induction-based argument. It can easily be shown that the induced kernel takes the form

$$K(\boldsymbol{x}, \boldsymbol{y}) = \Psi\left(\frac{\langle \boldsymbol{x}, \boldsymbol{y}\rangle}{\|\boldsymbol{x}\| \|\boldsymbol{y}\|}, \frac{\|\boldsymbol{x}\|^2}{d}, \frac{\|\boldsymbol{y}\|^2}{d}\right),$$

for some function $\Psi : \mathbb{R}^3 \to \mathbb{R}$. In addition, since $\boldsymbol{x}, \boldsymbol{y} \in \boxdot^d$, they have the same norm, and $\Psi$ can be treated as a univariate function that only depends on $c = d^{-1} \langle \boldsymbol{x}, \boldsymbol{y}\rangle$, *i.e.* $\Psi(c, 1, 1) = \Psi(c)$.

Using this property and the following result, it follows that the kernel induced by a transformer with linear attention is diagonalized by the Fourier basis $\{\chi_U\}_{U \subseteq [d]}$.

**Theorem B.2** (Theorem 3.2 in (Yang & Salman, 2020)). *On the $d$-dimensional boolean cube $\boxdot^d$, for every $U \subseteq [d]$, $\chi_U$ is an eigenfunction of $K$ with eigenvalue*

$$\mu_{|U|} := \mathop{\mathbb{E}}_{\boldsymbol{x} \sim \boxdot^d}\left[x^U K(\boldsymbol{x}, \boldsymbol{1})\right] = \mathop{\mathbb{E}}_{\boldsymbol{x} \sim \boxdot^d}\left[x^U \Psi\left(d^{-1} \sum_i x_i\right)\right],$$

*where $\boldsymbol{1} := (1, \dots, 1) \in \boxdot^d$. This definition of $\mu_{|U|}$ does not depend on the choice $S$, only on the cardinality of $S$. These are all of the eigenfunctions of $K$ by dimensionality considerations.*

Further, using the following result, it follows that transformers (with linear attention) exhibit weak spectral simplicity bias.

**Theorem B.3** (Theorem 4.1 in (Yang & Salman, 2020)). *Let $K$ be the CK or NTK of an MLP on a boolean cube $\boxdot^d$. Then the eigenvalues $\mu_k, k = 0, \dots, d$, satisfy*

$$\mu_0 \geq \mu_2 \geq \cdots \geq \mu_{2k} \geq \dots,$$
$$\mu_1 \geq \mu_3 \geq \cdots \geq \mu_{2k+1} \geq \dots.$$

## B.2 PROOF OF PROPOSITION 2.2

First, we introduce the concept of noise stability $Q_\rho(f)$, which measures the correlation between the outputs of a function $f$ for $\rho$-correlated pair $(\boldsymbol{x}, \boldsymbol{x}')$, as $Q_\rho(f) := \mathop{\mathbb{E}}_{(\boldsymbol{x}, \boldsymbol{x}')_\rho} f(\boldsymbol{x}) f(\boldsymbol{x}')$. Note that $Q_\rho(f)$ is related to $R_\rho(f) := \mathop{\Pr}_{(\boldsymbol{x}, \boldsymbol{x}')_\rho}[f(\boldsymbol{x}) \neq f(\boldsymbol{x}')]$ as $Q_\rho(f) = 1 - 2R_\rho(f)$ (O'Donnell, 2014).

Using the following result, we can relate noise stability to the Fourier weight of the function $f$ at different degrees $i \in [d]$.

**Theorem B.4** (Theorem 2.49 in (O'Donnell, 2014)). *For function* $f : \boxbar^d \to \{\pm 1\}$, *the noise stability for* $\rho$-*correlated pair* $(\boldsymbol{x}, \boldsymbol{x}')$ *satisfies* $Q_\rho(f) = \sum_{U \subseteq [d]} \rho^{|U|} \hat{f}(U)^2 = \sum_{i=0}^{d} \rho^i W^i[f]$, *where*
$W^i[f] := \sum_{U \subseteq [d], |U|=i} \hat{f}(U)^2.$

Clearly, $Q_\rho(f) \leq 1$ since the minimum degree of $f$ is 0. Next, we use the following important result, which upper bounds the degree of $f$ in terms of its maximum sensitivity, $S_{\max}(f) := \max_{\boldsymbol{x} \in \boxbar^d} S(f, \boldsymbol{x})$.

**Theorem B.5** (Theorem 1.4 in (Huang, 2019)). *For function* $f : \boxbar^d \to \{\pm 1\}$, *the degree* $D(f)$ *of the multilinear polynomial which represents* $f$ *satisfies* $D(f) \leq (S_{\max}(f))^2$.

Using this, and the fact that $Q_\rho(f)$ is minimized when the Fourier weight is concentrated on the highest degree term, we get the lower bound $Q_\rho(f) \geq \rho^{(S_{\max}(f))^2}$, since $\rho < 1$ and $S_{\max}(f) \in [1, d]$. Using the relation between $Q_\rho(f)$ and $R_\rho(f)$ then finishes the proof.

## C  RELATED WORK

**Understanding Transformers.** The emergence of transformers as the go-to architecture for many tasks has inspired extensive work on understanding the internal mechanisms of transformers, including reverse-engineering language models (Wang et al., 2022), the grokking phenomenon (Power et al., 2022; Nanda et al., 2023), manipulating attention maps (Hassid et al., 2022; Kobayashi et al., 2024), automated circuit finding (Conmy et al., 2023), arithmetic computations (Hanna et al., 2023; Quirke & Barez, 2024), optimal token selection (Tarzanagh et al., 2023a;b; Vasudeva et al., 2024), and in-context learning (Brown et al., 2020; Garg et al., 2022; Akyürek et al., 2023; von Oswald et al., 2022; Fu et al., 2023; Bhattamishra et al., 2023a; Guo et al., 2024). Several works investigate why vision transformers (ViTs) outperform CNNs (Trockman & Kolter, 2022a; Raghu et al., 2021; Melas-Kyriazi, 2021), as well as other properties of ViTs, such as robustness to (adversarial) perturbations and distribution shifts (Bai et al., 2023; Shao et al., 2021; Mahmood et al., 2021; Bhojanapalli et al., 2021; Naseer et al., 2021; Paul & Chen, 2022; Ghosal et al., 2022). Further, several works on mechanistic interpretability of transformers share a similar recipe of measuring sensitivity — corruption with Gaussian noise (Meng et al., 2022; Conmy et al., 2023) but on hidden states rather than the input space.

**Sensitivity and Spectral Bias.** Sensitivity is closely related to spectral bias (Yang & Salman, 2020), which is a bias towards 'simple' functions in the Fourier space. Simple functions in the Fourier space generally correspond to low-frequency terms when the input space is continuous, and low-degree polynomials when the input space is discrete. Recent work has shown that deep networks prefer to use low-frequency Fourier functions on images (Xu et al., 2019), and low-degree Fourier terms on Boolean functions (Yang & Salman, 2020). We note that in contrast to some other notions of spectral bias, sensitivity also has the advantage that it can be efficiently estimated on data through sampling — in contrast, estimating all the Fourier coefficients requires time exponential in the dimensionality of the data and hence can be computationally prohibitive (Xu et al., 2019).

**Simplicity Bias in DL.** Several works (Neyshabur et al., 2014; Valle-Perez et al., 2019; Arpit et al., 2017; Geirhos et al., 2020) show that NNs prefer learning 'simple' functions over the data. Nakkiran et al. (2019) show that during the early stages of SGD training, the predictions of NNs can be approximated well by linear models. Morwani et al. (2023) show that 1-hidden-layer NNs exhibit simplicity bias to rely on low-dimensional projections of the data, while (Huh et al., 2021) empirically show that deep NNs find solutions with lower rank embeddings. (Shah et al., 2020) create synthetic datasets where features that can be separated by predictors with fewer piece-wise linear components are considered simpler, and show that in the presence of simple and complex features with equal predictive power, NNs rely heavily on simple features. Geirhos et al. (2019) show that trained CNNs rely more on image textures rather than image shapes to make predictions. Rahaman et al. (2019a)

use Fourier analysis tools and show that deep networks are biased towards learning low-frequency functions, and (Xu et al., 2019; Cao et al., 2021; Bietti & Mairal, 2019; Basri et al., 2019) provide further theoretical and empirical evidence for this.

**Implicit Biases of Gradient Methods.** Several works study the implicit bias of gradient-based methods for linear predictors and MLPs. Pioneering work by Soudry et al. (2018); Ji & Telgarsky (2018) revealed that linear models trained with gradient descent to minimize an exponentially-tailed loss on linearly separable data converge (in direction) to the max-margin classifier. Following this, Nacson et al. (2019); Ji & Telgarsky (2021); Ji et al. (2021) derived fast convergence rates for gradient-based methods in this setting. Recent works show that MLPs trained with gradient flow/descent converge to a KKT point of the corresponding max-margin problem in the parameter space, in both finite (Ji & Telgarsky, 2020; Lyu & Li, 2020) and infinite width (Chizat & Bach, 2020) regimes. Further, Phuong & Lampert (2021); Frei et al. (2022); Kou et al. (2023) have also studied ReLU/Leaky-ReLU networks trained with gradient descent on nearly orthogonal data. Li et al. (2022) show that the training path in over-parameterized models can be interpreted as mirror descent applied to an alternative objective. In regression problems, when minimizing the mean squared error, the bias manifests in the form of rank minimization (Arora et al., 2019; Li et al., 2021). Additionally, the implicit bias of other optimization algorithms, such as stochastic gradient descent and adaptive methods, has also been explored in various studies (Blanc et al., 2020; HaoChen et al., 2021); see the recent survey (Vardi, 2022) for a detailed summary.

**Robustness.** Several research efforts have been made to investigate the robustness of Transformers. Shao et al. (2021) showed that Transformers exhibit greater resistance to adversarial attacks compared to other models. Additionally, Mahmood et al. (2021) highlighted the notably low transferability of adversarial examples between CNNs and ViTs. Subsequent research (Shen et al., 2023; Bhojanapalli et al., 2021; Paul & Chen, 2022) expanded this robustness examination to improve transformer-based language models. Shi et al. (2020) introduced the concept of robustness verification in Transformers. Various robust training methods have been suggested to enhance the robustness guarantees of models, often influenced by or stemming from their respective verification techniques. Shi et al. (2021) expedited the certified robust training process through the use of interval-bound propagation. Wang et al. (2021) employed randomized smoothing to train BERT, aiming to maximize its certified robust space. Recent work of Bombari & Mondelli (2024) shows that randomly-initialized attention layers tend to have higher word-level sensitivity than fully connected layers. In contrast to our work, they consider word sensitivity, which has been experimentally shown to be similar for transformers and LSTMs (Bhattamishra et al., 2023b).

**Spurious Correlations.** A common pitfall to the generalization of neural networks is the presence of spurious correlations (Sagawa et al., 2020). For example, Geirhos et al. (2019) observed that trained CNNs are biased towards textures rather than shapes to make predictions for object recognition tasks. Such biases make NNs vulnerable to adversarial attacks. Gururangan et al. (2018) attribute the reliance of NNs on spurious features to confounding factors in data collection while Shah et al. (2020) attribute it to a *simplicity bias*. Several works have studied the underlying causes of simplicity bias (Chiang, 2021; Nagarajan et al., 2021; Morwani et al., 2023; Huh et al., 2021; Lyu et al., 2021) and multiple methods have been developed to mitigate this bias and improve generalization (Pezeshki et al., 2020; Kirichenko et al., 2022; Vasudeva et al., 2023; Tiwari & Shenoy, 2023).

**Data Augmentation.** The essence of data augmentation is to impose some notion of regularization. The simplest design of data augmentation dates back to Robbins (1951) where image manipulation, e.g., flip, crop, and rotate, was introduced. Bishop (1995) proved that training with Gaussian noise is equivalent to Tikhonov regularization. We also note this observation is in parallel to our proposition in Section 6 that training with Gaussian noise promotes low sensitivity. Recently, mixup-based augmentation methods have been proposed to improve model robustness by merging two images as well as their labels (Zhang et al., 2018). Several works also use a combination of existing augmentation techniques (Cubuk et al., 2019; Lim et al., 2019). A common belief is that data augmentation can improve model robustness (Rebuffi et al., 2021), and this work bridges the method (augmentation) and the outcome (robustness) with an explanation — simplicity bias towards low sensitivity.

# D    DETAILS OF EXPERIMENTAL SETTINGS

We use PyTorch (Paszke et al., 2019) as our code framework and as our implementation of LSTMs. PyTorch is licensed under the Modified BSD license.

**Experimental Settings for Synthetic Data Experiments.** We use standard SGD training with batch size 100. We consider $T = 50$ and train with 1000 samples and test on 500 samples generated as per Definition 3.3. **Datasets, Model Architectures and Experimental Settings for Vision Tasks.** We consider the following datasets:

*Fashion-MNIST.* Fashion-MNIST (Xiao et al., 2017) consists of $28 \times 28$ grayscale images of Zalando's articles. This is a 10-class classification task with $60k$ training and $10k$ test images. It is released under the MIT license.

*CIFAR-10.* The CIFAR-10 dataset (Krizhevsky, 2009) is a well-known object recognition dataset. It consists of $32 \times 32$ color images in 10 classes, with $6k$ images per class. There are $50k$ training and $10k$ test images. It is released under the MIT license.

*SVHN.* Street View House Numbers (SVHN) (Netzer et al., 2011) is a real-world image dataset used as a digit classification benchmark. It contains $32 \times 32$ RGB images of printed digits (0 to 9) cropped from Google Street View images of house number plates. There are $60k$ images in the train set and $10k$ images in the test set. It is released under the CC BY 4.0 license.

*ImageNet-1k.* The ImageNet-1k dataset (Russakovsky et al., 2015), also known as the ILSVRC (ImageNet Large Scale Visual Recognition Challenge) dataset, is a widely used benchmark dataset in computer vision for tasks such as image classification, object detection, and image segmentation. There are 1000 different classes, and approximately 1.28 million training images of size $224 \times 224$. It is released under a non-commercial research use license.

For all the datasets, we use the ViT-small architecture implementation available at `https://github.com/lucidrains/vit-pytorch`. For the ResNet-18 model used in the experiments on CIFAR-10 and SVHN datasets, we use the implementation available at `https://github.com/kuangliu/pytorch-cifar`. Additionally, for the DenseNet-121 model, ConvMixer model and ViT-simple model used in the experiments on CIFAR-10, we use the implementations available at `https://github.com/huyvnphan/PyTorch_CIFAR10`, `https://github.com/locuslab/convmixer` and `https://github.com/lucidrains/vit-pytorch`, respectively. All of these models are released under the MIT license.

All the models are trained with SGD using batch size 50 for MNIST and 100 for the other datasets. We use patch size 7 for MNIST and 4 for the other datasets. We estimate the expectation over $\mathcal{P}$ in Definition 3.1 by replacing every patch with a noisy patch 5 times, and sample about $30\%$ of the training data to evaluate sensitivity.

For the MNIST experiments, we consider a 1-hidden-layer MLP with 100 hidden units and LeakyReLU activation. We set depth as 2, number of heads as 1 and the hidden units in the MLP as 128 for the ViT. We train both models with a learning rate of 0.01.

For Fashion-MNIST, we set depth as 2, number of heads as 8 and the hidden units in the MLP as 256 for the ViT. We consider a 2-hidden layer MLP with 512 and 128 hidden units, respectively. The CNN consists of two 2D convolutional layers with 32 output channels and kernel size 3 followed by a 2D MaxPool layer with both kernel size and stride as 2 and two fully connected layers with 128 hidden units. We use LeakyReLU activation for the CNN. We use learning rates of 0.1 for the MLP with LeakyReLU, 0.5 for the MLP with sigmoid, 0.005 for the CNN and 0.1 for the ViT.

For CIFAR-10, we set depth as 8, number of heads as 32 and the hidden units in the MLP as 256 for the ViT-small model, while these values are set as 6, 16, and 512 for the ViT-simple model. For the ConvMixer model, we use depth 6, embedding dimension 128 and kernel size 3. The learning rate is set as 0.1 for ViT-small, 0.2 for ViT-simple, 0.06 for ConvMixer, 0.001 for ResNet-18 and 0.005 for DenseNet-121.

For SVHN, most of the settings are the same as the CIFAR-10 experiments, except we set the hidden units in the MLP as 512 for the ViT-small model and the learning rate is set as $0.0015$ for ResNet-18.

For ImageNet-1k, we sample $20k$ samples from the training set and compare the sensitivity values of pre-trained ConvNext (`ConvNextV2-Tiny`) and ViT/L-16 with $\sigma^2 = 15$. Both achieved the same training and validation accuracy of $85\%$, and it ensures our sensitivity comparison is fair.

**Datasets, Model Architectures and Experimental Settings for Language Tasks.** We consider the following two binary classification datasets, which are relatively easy to learn without pretraining (Kovaleva et al., 2019).

*MRPC.* Microsoft Research Paraphrase Corpus (MRPC) (Dolan & Brockett, 2005) is a corpus that consists of 5801 sentence pairs. Each pair is labeled if it is a paraphrase or not by human annotators. It has 4076 training examples and 1725 validation examples. It is released under the ODC-By or the Microsoft Research license.

*QQP.* Quora Question Pairs (QQP) (Iyer et al., 2017) dataset is a corpus that consists of over $400k$ question pairs. Each question pair is annotated with a binary value indicating whether the two questions are paraphrases of each other. It has $364k$ training examples and $40k$ validation examples. It is released under the CC BY-SA 2.5 license.

For both RoBERTa models and LSTM models, we keep the same number of layers: 4 layers. We set number of heads as 8 RoBERTa. We use the AdamW optimizer with a learning rate of $0.0001$ and weight decay of $0.0001$ for all the tasks. We also use a dropout rate of $0.1$. We use a batch size of 32 for all the experiments. The used RoBERTa model is released on Huggingface https://huggingface.co/FacebookAI/roberta-base with MIT license.

**Experimental Settings for Section 6.** We set the learning rate as $0.16$ and $0.2$ when training the ViT-small with regularization and augmentation, respectively. We use a regularization strength of $0.25$. The remaining settings are the same as for the other experiments. For computing the sharpness metrics, we approximate the expectation over the Gaussian noise by averaging over 5 repeats and set $\sigma$ as $0.005$.

**Experimental Settings for Appendix A.3.** For the experiment with the Adam optimizer, we employ a learning rate scheduler to ensure that the accuracy on the train set is similar to the model trained with SGD. The initial learning rate is $0.002$ and after every 8 epochs, it is scaled by a factor of $0.5$.

For the remaining experiments in this section, we consider the same settings as for the respective main experiment.

**Compute Details.** Experiments with synthetic data were run on Google Colab. Experiments on vision and language tasks were run on internal clusters using NVIDIA RTX A6000 GPUs with 48GB of VRAM. For the experiments on vision data, we use two GPUs and the runtime for each setting is about 17 hours. Experiments on language tasks use one GPU and the runtime for each experiment is about 24 hours.

# E    LIMITATIONS

In our theoretical results, we show that transformers exhibit weak spectral bias, similar to other NN architectures. An important direction for future work is to distinguish transformers from other architectures and show that they exhibit a stronger spectral bias.

Additionally, this work focuses on the inductive bias of the transformer architecture. However, other factors such as the data used while pre-training can also effect the biases these models exhibit on downstream tasks. It would be interesting to explore this effect in the future.

Similarly, the choice of the optimization algorithm used for training can also have an effect. In our experiments on the CIFAR-10 dataset (Fig. 11), we see that SGD and Adam are very similar. However, conducting a more thorough comparison, e.g., by considering second-order optimization methods, can be an important direction for future work.

