# OpenReview forum: "Transformers Learn Low Sensitivity Functions: Investigations and Implications"
_ICLR.cc/2025/Conference — ICLR 2025 Poster_

### Official Review · Reviewer_eYcM · 2024-10-30

**Soundness:** 2
**Presentation:** 3
**Contribution:** 2
**Rating:** 6
**Confidence:** 4

**Summary:**

This work studies the sentitivity of functions defined by different deep learning architectures, comparing the specific case of Transformers with CNNs and Mixers. The work stems from previous work tha has studied sensitivity with Boolean inputs, and derives a formulation for token-based models. The authors make a connection between sensitivity and robustness, show how ViTs are less sensitive than other architectures and also show how sensitivity can be used for grokking analysis.
Experiments on synthetic data are provided, as well as experiments using ViT on small datasets (CIFAR, SVHN, ImageNet) and LLMs on 2 datasets (MRPC and QQP).

**Strengths:**

**Originality:**

Focusing on sensitivity starting from the Boolean formulation is original. I also found the experiment on a synthetic vocabulary (3.1) original.

**Clarity:**

The paper is well written, with clear language. The mathematical notation and formulation is also easy to read.

**Significance:**

The study of sensitivity in current models is important for interpretability as well as to design better training strategies.

**Weaknesses:**

**Originality:**

While sensitivity study has its originality, many previous works have studied sensitivity in many ways, for example by understanding the effects of image augmentations (see contrastive learning literature).

**Quality:**

The experiments provided are either synthetic or use small models/datasets. This makes the claims in the paper weaker in my opinion. For example:
* Results in Section 3 use synthetic data and a single attention layer. I would argue that, while still interesting, these experiments might not transfer to full models with several layers and multiple attention heads.
  * Related to this experiment, other research has been carried out analyzing spurious correlations. For example, the work by Robert Geirhos (among others) has already shown that CNNs tend to learn from the easiest cues available. In the experiments in Section 3.1, these "easy" cues would be the sparse tokens. Once they become uninformative, the next available (but harder) cue are the frequent tokens.

> Geirhos, Robert, et al. "Shortcut learning in deep neural networks." Nature Machine Intelligence 2.11 (2020): 665-673.

> Geirhos, Robert, et al. "ImageNet-trained CNNs are biased towards texture; increasing shape bias improves accuracy and robustness." arXiv preprint arXiv:1811.12231 (2018).

* Results in Section 4 use small datasets (CIFAR, SVHN) and arguably a medium size dataset nowadays (ImageNet). The models used (Vit-simple/small) are far from real scenarios nowadays, and the compared architectures are also small (3-layer CNN in for example).

* Results in Section 5 use a Roberta model (2019) which does not have the same properties as current LLMs. Also, this model is trained from scratch on small tasks, which also does not transfer to current abilities of LLMs.

In several cases, bold conclusions are extracted from a single model / single dataset experiment, with which I cannot agree. For example, the claim in L357 *_"Thus, transformers learn lower sensitivity functions compared to MLPs, ConvMixers, and CNNs"_* is validated with a 3-layer CNN on a small dataset like SVHN.

**Clarity:**

* It is not clear how the noising strategy is performed. The text mentions that _tokens_ are polluted with noise, however Fig 1 shows the noise applied to the pixel patch and says *_" the original image and the corrupted image are fed into the same neural network"_* (which implies that noise is applied at pixel level). The authors should clarify this important aspect.

* It is also not clear how noising is applied to CNNs (which are not patch/token based).

* Proposition 2.1 is harder to parse than the rest of the text, and it is hard to understand why it is important for the paper.

**Significance:**

While the objective of the paper is significant, the results provided and the size of the experiments laregly diminish the impact of this work.

**Questions:**

* Following up on my previous comment, the authors should clarify if the noising procedure is applied on patches (pixels) or token representations. Fig. 1 contradicts the text.
  * Also, how is noising applied on CNNs?

* How is $\sigma$ important, and why were different $\sigma$ chosen for the experiments in Section 4. I personally find it a drawback that one needs to find a right $\sigma$, and that using different ones the conclusions might change. Also, bold claims are provided with a single (different) sigma per dataset, which raises some questions.

* ViT/LLMs might produce different token scales, but $\sigma$ is kept fixed. This can impact strongly some tokens and leave others almost noise-less. I find this also a negative point of this algorithm, since some "topics" might by-pass the noising.

* How is a single attention representative of a large Transfomer in Section 3.1. I would ask the authors to elaborate on this.
  * Additionally, why have a linear layer $U$ after another linear layer $W_v$, since the composition of both is already a linear layer?

* In Fig. 4, only the training accuracy is provided. What about the test accuracies? It is known that models achieve perfect train accuracy, but the test accuracy might be very different though. Does the test accuracy correlate with the sensitivity (measured on train data as you already do)?

* About the claim in L347 *_"This shows that the observations on small-scale models studied in this section transfer to large-scale pretrained models."_*. By increasing scale, the sensitivity of a conv model has gone down to 0.0342, which is much lower than 0.0829 for ResNet-18. Also, ViT went up from 0.0014 to 0.0191. It would be fair to conclude that scaling up brings sensitivities closer, which would mean that small-scale does not transfer to large-scale. Also, one could go much larger in scale (larger ViT, larger datasets) and see if the trend is still maintained or sensitivities are even closer.

* Claims in Section 5 are obtained with one Language model (Roberta) and one LSTM on 2 small datasets. I cannot agree with the claims being generic for *Language Tasks* with this setup. Moreover knowing that current LLMs have much different properties than LMs used in 2019 (Roberta).

---

> ### Author Response · Authors · 2024-11-22
>
> Thank you for the detailed comments and feedback to help improve our work. We hope that the following responses will address the reviewer’s concerns and would be happy to address further questions the reviewer may have.
>
> **Weaknesses:**
>
> **1: Regarding the scale of experiments**
>
> *a. “Results in Section 3 use a single attention layer model”*
>
> We emphasize that the goal of Section 3 is to show that even in a very simple setting with a single-layer self-attention model, we can see the low sensitivity simplicity bias. While the dataset **can** be considered for analyzing more complex model architectures as well, we choose not to do so because a) it is not necessary to use a more complex model for this data and b) it is more interesting to analyze larger models on more realistic settings. Hence, we compare larger models on real-world vision and language datasets in Sections 4 and 5. That being said, the experiments in Section 3 provide some insight into what part of the model gives rise to the low sensitivity bias.
>
> *b. “Experiments in Sections 4 and 5 use small-scale datasets and models”*
>
> We emphasize three points to address this concern.
>
> First, we respectfully disagree with the statement that “bold conclusions are extracted from a single model/dataset”. We compare several datasets and models to validate our claims. Specifically, to show that transformers learn lower sensitivity functions than CNNs, we consider two datasets as follows.
> - We consider the CIFAR10 dataset and compare two CNN-based models, namely ResNet18 and DenseNet12, with a ConvMixer model and and two ViT models, and
> - We compare a CNN (ResNet18) and a ViT on the SVHN dataset.
>
> Next, to show that transformers learn lower sensitivity functions than MLPs,
> - we consider the FashionMNIST dataset and compare a ViT, a simpler 3-layer CNN and MLP-based models with two activation functions.
> - we also consider a binary classification task with the MNIST dataset and compare a ViT and an MLP.
>
> We consider simpler models and datasets for comparison with MLPs because we expect these models to perform reasonably well on these datasets.
>
> Similarly, for comparisons with LSTMs, we consider two datasets MRPC and QQP, and compare with two language models, namely RoBERTa (in the main body) and GPT-2 (in Fig. 20 the Appendix, as mentioned in line 401 in the paper).
>
> Second, we note that we focus on relatively simpler datasets and models because we can train the models from scratch in these settings to get reasonably good performance. This is important because we compare the sensitivity of models that have comparable and reasonably good train accuracy. We want to single out the effect of the architecture, without any confounders such as the choice of the optimization algorithm or data augmentation or use of any pretraining strategies, which might vary for different architectures to get good accuracy.
>
> That being said, we compare (pre-trained) ConvNeXT and VIT-B/16 models on ImageNet-1K dataset to show that the conclusions are relatively robust to pretraining and increasing scale. However, a systematic and fairer comparison for large-scale models would warrant better control over the pretraining strategies, which is beyond the scope of this work.
>
> Third, we acknowledge the reviewer’s concern about the claims not directly transferring to LLMs. However, we emphasize that our work aims to give insights about the inductive biases and other properties of the transformer architecture in general, and we don’t claim that these insights transfer directly to LLMs or about the interactions with pretraining and finetuning. That being said, comparisons with GPT-2 model, which is a causal model and more similar to recently used language models compared to RoBERTa, lead to the same conclusions and indicate that we can expect them to also hold for other language models.
>
>
> **2: Clarity**
>
> *a. “It is not clear how noising is applied to image data”*
>
> Fig.1 is exactly what is done for images – adding noise to one patch at a time. Since noise is at the patch level, the process is the same for CNNs and ViTs.
>
> *b. “It is hard to understand the importance of Proposition 2.1”*
>
> As stated in lines 154-157 in the paper, “Larger eigenvalues for lower-order monomials indicate that simpler features are learned faster. Since low sensitivity implies learning low-degree polynomials, Proposition 2.1 also implies a weak form of low sensitivity bias.”

---

> > ### Comment · Reviewer_kdeM · 2024-11-24
> >
> > I have read the reviewer's concerns and the authors' response. Since this review has given a score that I found surprisingly low for this paper, I take the liberty to bring to the reviewer's attention that the authors seem to have addressed the reviewer's concerns. In particular, I agree with the authors that the noisy strategy is described and also that lines 154-157 clarify the implications of Proposition 2.1.
> >
> > Regarding the limitations of the experiments and the datasets, it is my understanding that this paper explores the sensitivity and inductive bias of transformers, and present rigorous theoretical arguments. The experiments seem tailored to the main aims of the paper. Respectfully, why does the reviewer find it necessary to provide additional experiments on more datasets and models, considering the main goal of the paper? Of course, more experiments on models and datasets with more practical relevance are helpful, but is lack of such experiments enough grounds for rejection? On the same note, doesn't the reviewer think this very same objection could hold for a large set of accepted papers at ICLR and other venues, many of which make valuable contributions?

---

> > > ### Comment · Reviewer_eYcM · 2024-11-28
> > > **Answer to reviewer kdeM in the review thread**
> > >
> > > I appreciate your enthusiasm about this work being accepted. However, I had several concerns that required clarification, which is common during a review process, since we all have different opinions and reviewing critera.
> > >
> > > I would be happy to discuss this further during the reviewer-reviewer discussion period if needed.

---

> ### Author Response · Authors · 2024-11-22
>
> **Questions:**
>
> **Q1:** Please see the response to weakness 2a.
>
> **Q2: How does the variance $\sigma$ impact the results?**
>
> Thank you for the question. We evaluate the sensitivity values of the five models compared on the CIFAR10 dataset at the end of training using different values of $\sigma$. We added the results in Appendix A.3 and we see that even though the sensitivity values are different for different $\sigma$ (as expected), the conclusion that transformers learn lower sensitivity functions than CNNs is robust to the value of $\sigma$. Appendix A.4 also has results for the QQP dataset with a different value of $\sigma$ than considered in the paper.
>
> **Q3: Using fixed noise level for different tokens.**
>
> We believe that token norms are comparable and thus, it makes sense to use a fixed noise level. Specifically, noise is added at patch level for images, and as mentioned in line 377 in the paper, for language tasks, noise is added after layer normalization.
>
> **Q4: Attention architecture considered in Section 3.1**
>
> The model considered in Section 3.1 is a standard attention layer with key, query, and value weights $W_K, W_Q, W_V$, composed with a linear decoder $U$. Please also see the response to weakness 1a.
>
> **Q5: Does test accuracy correlate with sensitivity?**
>
> Fig 13 in the Appendix shows the test accuracies for the models considered in Fig. 4 on the CIFAR-10 dataset. Although the ViTs have a slightly higher test accuracy, it is comparable across the five models.
>
> **Q6: How does model scale affect the difference between sensitivity values?**
>
> This is an interesting question. However, as mentioned in the response to weakness 1b, comparing larger models is challenging because we also have to control the pretraining data and strategies for a fair comparison.
>
> **Q7: “Section 5 only considers the RoBERTa model. Do the claims hold for more recent language models?”**
>
> As mentioned in line 401 in the paper, we include the results with GPT-2 in the Appendix which lead to the same conclusions. Please also see the response to weakness 1b.

---

> > ### Comment · Reviewer_eYcM · 2024-11-27
> > **Answer to rebuttal**
> >
> > I would like to thank the authors for the details provided in their rebuttal. Most of my questions have been resolved after reading the rebuttal:
> >
> > * The experiment and discussion about $\sigma$ showing that _"even though the sensitivity values are different for different $\sigma$ (as expected), the conclusion that transformers learn lower sensitivity functions than CNNs is robust to the value of $\sigma$"_ is relevant.
> >
> > * The discussion and clarification about noise being added at pixel level for images, and after LN for text. This detail is important and I might have missed it during my initial review, assuming that _token_ referred to some representation of a piece of data (either image or text). In such case, I agree in that ranges are consistent across images (or text if LN is used) and that a fixed noise level is a reasonable choice.
> >   * I wonder how your approach works if applied on LN of image models? I don't see why it would not work, but it might be interesting to show for consistency with the text modality.
> >
> > * The overall response to the _simplicity_ of the model in Section 3, summarized as _"the goal of Section 3 is to show that even in a very simple setting with a single-layer self-attention model, we can see the low sensitivity simplicity bias"_. I believe this could be stated upfront in that section, so the reader understands the focus. As it is now, in L172 it is stated _"In order to investigate the inductive biases in real-world image and language tasks, we need an equivalent metric for high-dimensional, real-valued data"_. I suggest explaining that a simple setup as proof-of-concept will be provided in Sec 3.1, and that real-world scenarios will be provided in Sec 4, 5.
> >
> > * I believe the overall focus of the paper could be emphasized, as the authors have done in their answer. For example, the comment: _"we emphasize that our work aims to give insights about the inductive biases and other properties of the transformer architecture in general, and we don’t claim that these insights transfer directly to LLMs or about the interactions with pretraining and finetuning."_ is a good example. Some direct statement like this could also help the reader to understand the contributions and limitations of this work.
> >
> > **Still some questions:**
> >
> > * Noising strategy:
> >
> > > _Since noise is at the patch level, the process is the same for CNNs and ViTs._
> >
> > If I understand correctly, for CNNs, noise is added on an image patch (a bounding box within the full image). However, CNNs do not consume patches, but rather _scan_ the whole image with convolutional kernels. In my opinion, adding Gaussian noise to a specific part of the image has not the same effect as adding noise to a ViT token. Could the authors comment on that aspect?
> > Why was pixel noising preferred over noising at LN level as done in the text case?
> >
> > * Connection with [1, 2]. The comment provided in my review:
> >
> > > _CNNs tend to learn from the easiest cues available. In the experiments in Section 3.1, these "easy" cues would be the sparse tokens. Once they become uninformative, the next available (but harder) cue are the frequent tokens._
> >
> > is something that I think should be addressed, since the connection with [1, 2] might reduce novelty. This is somehow related with the concern raised by Reviewer A7Md about the connection with works relating robustness and augmentations.
> >
> > > Geirhos, Robert, et al. "Shortcut learning in deep neural networks." Nature Machine Intelligence 2.11 (2020): 665-673.
> >
> > > Geirhos, Robert, et al. "ImageNet-trained CNNs are biased towards texture; increasing shape bias improves accuracy and robustness." arXiv preprint arXiv:1811.12231 (2018).
> >
> > **Overall comment:**
> >
> > The authors have provided thorough responses and clarification that have improved my confidence in this work, as well as in the experimental setup. However, there are some aspects I consider worth clarifying before making a decision about acceptance.

---

> > > ### Author Response · Authors · 2024-11-28
> > >
> > > We thank the reviewer for discussing with us and we are glad that most of the reviewer’s concerns were resolved. We would like to further address the reviewer's questions and comments as follows.
> > >
> > > **Why add noise to image patches?**
> > >
> > > To ensure fair comparisons of sensitivity between CNNs and ViTs, we want to ensure that the input images are corrupted in the same way and to adhere to the definition of sensitivity, only some local tokens are corrupted. This is why we chose to corrupt image patches with Gaussian noise. Although CNNs are designed to convolve the images, their convolution kernels are usually small and local, for example, the kernel size of ResNet is only 3x3. In this sense, at early layers, CNNs still process local pixels and this will further contribute to their overall sensitivity.
> > >
> > >
> > > **Why not add noise after LayerNorm?**
> > >
> > > Good question! The goal of our experiments is to compare the sensitivity of ViT with that of many architectures such as ResNet, DenseNet, MLP, and ConvMixers. Unfortunately, most of these architectures use BatchNorm instead of LayerNorms. It would be nice if we modify their architectures to LayerNorms but that seems beyond the scope of the main message of this paper.
> > >
> > >
> > > **Connection to prior work**
> > >
> > > We agree with the reviewer that there are connections with [1,2]. We discuss related work on simplicity bias in deep learning in lines 1449-1460 in the Appendix (due to space constraints). It is evident from prior work that the notion of simplicity can vary from one architecture to another while characterizing the simplicity bias. For instance, CNNs trained for object recognition tasks have been found to rely on texture rather than shape to make predictions, while MLPs (specifically, 1 hidden-layer NNs) have been found to rely on a lower-dimensional projection of the input data to make predictions.
> > >
> > > One of the contributions of our work is to identify the metric that can distinguish between transformers and other architectures. Our work identifies low sensitivity as a notion of simplicity bias for transformers, that is observed systematically across various settings. We also note that it has several useful properties, as discussed in the paper, like being a natural analog of the notion of sensitivity used in Boolean function analysis, being predictive of properties like robustness and generalization, and serving as a progress measure for grokking.
> > >
> > >
> > > Again, we appreciate the reviewer’s time and engagement in the discussion and we would be happy to answer any further questions or concerns.

---

> > > > ### Comment · Reviewer_eYcM · 2024-11-28
> > > >
> > > > Thanks for your last message with an interesting discussion about the topics raised. I find the answers sensible, and I encourage the authors to include such comments in the final paper, in the form the authors find more reasonable.
> > > >
> > > > Given the authors' engagement, grounded answers, and the much more clear focus of the paper and setup, I will update my score above acceptance.

---

> > > > > ### Author Response · Authors · 2024-11-28
> > > > >
> > > > > Thank you for your thoughtful feedback and for updating your score in support of our paper. We appreciate your time and effort and will incorporate your suggestions into the final version.

---

### Official Review · Reviewer_kdeM · 2024-11-01

**Soundness:** 4
**Presentation:** 4
**Contribution:** 3
**Rating:** 8
**Confidence:** 3

**Summary:**

The paper explores the inductive biases of transformers, particularly their tendency to learn low-sensitivity functions. It introduces sensitivity as a measure of how model predictions respond to token-wise perturbations in the input. By comparing transformers to other architectures like MLPs, CNNs, ConvMixers, and LSTMs across both vision and language tasks, the paper shows that transformers consistently exhibit lower sensitivity. This low-sensitivity bias is linked to improved robustness, flatter minima in the loss landscape, and hence better generalization. Additionally, the authors propose that sensitivity could act as a progress measure in training, and is linked to grokking.

**Strengths:**

**Key strength:** In addition to the general importance of developing rigorous understanding of how transformers work and why they show such remarkable properties, this paper proposes a novel perspective by looking into sensitivity. They rigorously define sensitivity and provide strong arguments on how it links to other important properties, such as robustness and generalization. They also show that it can track progress when grokking happens, which I think is an important finding and could potentially enable a series of future studies on grokking.

**Other strengths:** Here are a list of other points that I commend the authors for:
- The introduction is quite well written and motivates the main question quite well (thought it could be improved; see weaknesses). Similarly, the contributions are well explained at the end of the introduction.
- The presentation of the paper is strong, and maintains a good balance between accessibility and rigor.
- Propositions 2.1. And 2.2 are really interesting results on the spectral bias and sensitivity of transformers.
- The authors explain the implications of their theory quite well.
- The experimental design is thorough and well-tailored to validating the theory.
- While I consider this a theoretical paper, the experiments are quite strong and cover various aspects of the paper’s main questions.

**Weaknesses:**

I do not see any major weakness. But there could be some improvements. See my suggestions for improvement, below.
1. While the paper clearly explains that lower sensitivity is linked to higher robustness, trade-off/connection with expressivity and performance are not discussed. There is a well-established trade-off in various contexts (see, e.g., [1-2]), and it would further strengthen the paper to discuss this.

2. Though I think the introduction is quite well-written, I think it under-emphasizes the findings of the paper on the role of sensitivity analysis. The authors conduct a rigorous analysis of the transformers sensitivity and use that to clarifies some of the important properties of transformers as I mentioned for the strengths, but while doing so, they also show, quite rigorously with strong theory and experiments, how sensitivity analysis could be used to understand generalization, grokking, etc. Near the end of the paper this realization caught my attention, and the authors actually do point this out more clearly in the Conclusion, but I think this can be better emphasized in the Introduction.

3. I suggest the authors bring the Limitation section from the appendix to the main paper. The limitations are not discussed in the main paper, while it is always important to discuss them.

4. This is a rather minor point and it might be a matter of taste: Do sections 5 and 6 really need to be separate sections? It seems like the findings are generally similar, and they could be merged in one section of empirical analysis of vision and language models.


**References**

[1] Zhang, H., Yu, Y., Jiao, J., Xing, E., El Ghaoui, L., & Jordan, M. (2019, May). Theoretically principled trade-off between robustness and accuracy. In International conference on machine learning (pp. 7472-7482). PMLR.

[2] Raghunathan, A., Xie, S. M., Yang, F., Duchi, J., & Liang, P. (2020). Understanding and mitigating the tradeoff between robustness and accuracy. arXiv preprint arXiv:2002.10716.

**Questions:**

1. Related to weakness 1, how do you think the sensitivity and robustness relate to expressivity and performance in transformers?

2. Lines 310-311 mention generalization capabilities of different models as a reason to investigate sensitivity during training. This made me curious, what do you think the connection between generalizability in representation learning or classification relates to generalizability in sensitivity (I think one direction of it is clear, but the other direction is not)?

3. In line 394, you mention that use the same number of layers for LSTM and RoBERTa for fair comparison. How about the model size in terms of number of parameters? How many parameters in each model? And how do you think changing this could impact your results?

**Details Of Ethics Concerns:**

No ethics concerns.

---

> ### Author Response · Authors · 2024-11-22
>
> Thank you for the positive feedback and the helpful comments.
>
>
> **W1, Q1: How do sensitivity and robustness relate to the tradeoff between robustness and accuracy observed in different contexts?**
>
> Thank you for the question. We note that these tradeoffs occur when measuring robustness to adversarial perturbations or out-of-distribution (OOD) data. In this paper, we focus on benign shifts to measure robustness and hence, don’t observe this type of a tradeoff.
>
>
> **W2-4**: Thank you for the suggestions to improve the presentation of the paper. We have made some edits to the introduction as suggested. We will also incorporate the other suggestions in the final version.
>
>
> **Q2: Relation between generalization in representation learning to generalization in sensitivity**
>
> This is an interesting question. We believe that under benign shifts, such as on new samples from the same distribution as the train set, or under small random corruptions, most properties should generalize. At the very least, the conclusions drawn based on the metrics should generalize, even if the values change. Exploring the generalization of sensitivity systematically can be an interesting direction for future work.
>
>
> **Q3: Comparing LSTM and RoBERTa with the same number of parameters.**
>
> As mentioned in the paper, we compare models that have the same accuracy to ensure the comparison of sensitivity values is fair. Since both the models attain very similar accuracy, they can potentially learn similar functions that could attain similar sensitivity values. However, we observe that they learn functions that have similar accuracy but differ significantly in terms of sensitivity.

---

> > ### Comment · Reviewer_kdeM · 2024-11-24
> >
> > Thank you addressing my concerns and answering my questions. I maintain my score and I would like to take the chance to again commend the authors for the merits of their research and writing this strong paper. This is strong paper, my assessment is that the quality of this paper is substantially higher than a typical ICLR paper (based on those I have read from previous years, of course), and **I strongly recommend acceptance of this paper for publication at ICLR**.

---

> > > ### Author Response · Authors · 2024-11-28
> > >
> > > We sincerely thank the reviewer for their positive feedback and suggestions that helped improve our work. We greatly appreciate your support and encouragement.

---

### Official Review · Reviewer_A7Md · 2024-11-06

**Soundness:** 1
**Presentation:** 3
**Contribution:** 3
**Rating:** 5
**Confidence:** 3

**Summary:**

This work builds on the theoretical notion of boolean sensitivity, extending it to an empirically measurable quantity and studying it for the case of transformers. It finds that transformers have lower input sensitivity on the training data, compared to other architectures, and that this is correlated with other phenomena such as test-time robustness, sharpness of minima, and grokking.

**Strengths:**

The study in the paper is quite intriguing. A few things I liked:
 * Provides a new lens on what is different about transformers
 * Demonstrates phenomena consistently across many datasets
 * Provides a new lens on grokking not captured by the weight norm

**Weaknesses:**

There were a few key places where I felt the paper overclaimed or made dubious claims, which are enough for me to not favor acceptance. In particular:
 * Lower sensitivity leads to robustness: this is basically a restatement of the claim that Gaussian data augmentation improves robustness. This is a very well-known result, the authors do say that it is in line with other results in the literature, but I feel they are understating the extent to which this is well-trodden ground (for instance, Hendrycks, one of the authors of CIFAR-10-C, has published an entire line of work on data augmentation methods for improving robustness; Gaussian noise is the simplest of these and many others work far better).
 * Perhaps more importantly, this sentence does not seem merited: "Together, these results indicate that the inductive bias of transformers to learn functions of lower sensitivity *explains* the improved robustness (to common corruptions) compared to CNNs." I am not sure what "explains" means, but there are many other interventions that improve robustness (such as the data augmentation methods mentioned above), and some of those might have better explanatory power.
 * It is not entirely clear whether input sensitivity is a *different* phenomena than test-time robustness to perturbations. The main difference is it is computed on the training set instead of the test set --- but are there cases where these come apart, or is test-time and train-time input sensitivity always highly correlated?
    * I think the results could be interesting either way -- but if they are the same, then this is interesting mainly because it is a proxy for robustness that can be computed at training time; if they are different, then understanding the differences would be interesting.

**Questions:**

Have you compared input sensitivity and perturbation robustness at test time? When if ever do they behave differently?

---

> ### Author Response · Authors · 2024-11-22
>
> Thank you for the helpful comments and feedback.
>
>
> **W1-2: Regarding the claim that low sensitivity explains the better robustness of transformers.**
>
> Following the reviewer’s suggestion, we have rephrased the aforementioned statement in the paper as follows: “As encouraging lower sensitivity improves robustness, the inductive bias of transformers to learn functions of lower sensitivity could explain their better robustness
> (to common corruptions) compared to CNNs.”
>
> As the reviewer mentioned, prior works have shown that transformers are more robust in CNNs. We note that we discuss related work on robustness and data augmentation in lines 1467-1479 and lines 1489-1499, respectively in the Appendix. However, discussing the difference in the robustness of transformers and CNNs in more detail helps us elucidate the connection between lower sensitivity and better robustness.
>
> We summarize our results from Section 6.1 below and hope our response and the revised statement address the reviewer’s concern about the section.
>
> In Sections 4 and 5, we showed that transformers have lower sensitivity compared to other architectures, and in Section 6.1, we investigate the role of lower sensitivity in the improved robustness of transformers. Using the CIFAR-10-C dataset, we (a) observe that transformers have better robustness compared to CNNs, and (b) show that encouraging lower sensitivity while training the transformer further improves the robustness. Here, we emphasize three points.
>
> First, transformers exhibit a bias towards learning low-sensitivity functions even when trained without explicit regularization, as shown in the experiments in Sections 4 and 5.
>
> Second, our goal while training with data augmentation (with Gaussian noise added to the images randomly) and sensitivity regularization (with patch-wise Gaussian noise added to the images) is to see if we can show that reducing sensitivity leads to improved robustness. These methods seem like the simplest ways to encourage low sensitivity. We welcome any suggestions the reviewer may have about other ways we can encourage lower sensitivity, and will try our best to test those.
>
> Third, while one may expect that encouraging lower sensitivity while training would improve robustness to noise corruption, we observe improved robustness to other corruptions as well. For instance, corruptions from the blur, weather and digital transform categories are significantly different from the noise corruptions as they cannot be implemented by making small pixel-wise changes to the image. The improved robustness across these perturbations suggests that it could be a consequence of the lower sensitivity.
>
> In other words, while the definition of sensitivity is slightly similar to the corruptions from the noise category, it is quite different from the other corruptions and still correlates well with the improved performance on those.
>
>
> **W3, Q1: Comparing (train) sensitivity to perturbation robustness at test time.**
>
> As suggested by the reviewer, there could be connections between sensitivity and robustness to random perturbations, however, sensitivity as a metric not only serves as a measure of inductive bias that distinguishes transformers from other architectures but also has important implications, as a measure of robustness and flatness of the minimum and as a progress measure for grokking.
>
> For instance, in App. A.2, we evaluate sensitivity to Gaussian noise added across the input and observe that while transformers have lower sensitivity (or better robustness), this metric does not distinguish transformers from other architectures as clearly as the sensitivity to token-wise Gaussian perturbations. Similarly, some of the noise corruptions considered in the experiments in Section 6.1 also indicate that lower sensitivity is correlated with better robustness.
>
> That being said since we consider sensitivity a notion of inductive bias, it makes sense to measure it on the train test, analogous to other metrics of inductive bias such as maximum $\ell_2$-margin for linear predictors on separable data. This allows leveraging a property of the train data to predict things like generalization on the test data.

---

> > ### Comment · Reviewer_A7Md · 2024-11-24
> >
> > Thank you for your response. I appreciate the authors revising the claim about explanation. My other concerns remain, so I will keep my score.

---

### Official Review · Reviewer_pCqz · 2024-11-08

**Soundness:** 3
**Presentation:** 3
**Contribution:** 3
**Rating:** 8
**Confidence:** 2

**Summary:**

The paper considers implicit biases for the transformer architecture. They describe sensitivity of a function as the change in the function value averaged over all possible element-wise changes to the function input, averaged or maxed over all inputs on a hypercube. Such functions can be described using polynomials with the sensitivity connected to the degree of the polynomial. The paper proves that a linear attention transformer is biased (in the eigenvalue of the NTK sense) toward low-sensitivity functions characterized by the degree. Then they go on to generalize the notion of sensitivity for neighborhoods of general (non-boolean) inputs.

**Strengths:**

- Useful formalization of sensitivity
- Interesting findings about low sensitivity, robustness, and sensitivity to different parts of the input (like last token in a sequence)
- variety of tasks for better understanding of where architecture properties come from
- connecting between grokking and sensitivity provides a new lens into understanding and improving DNN training.

**Weaknesses:**

- On line 141, "where the eigenvalues are non-decreasing with the degree of the multi-linear monomials" would be easier if it said "eigenvalues do not decrease as the degree increases."

**Questions:**

- What's the difference between sensitivity and adversarial/robustness that looks at neighborhoods?

---

> ### Author Response · Authors · 2024-11-22
>
> Thank you for the positive feedback and the helpful comments.
>
>
> **W1**: Thank you for the suggestion, we have revised the sentence.
>
>
> **Q1: How does sensitivity relate to robustness to adversarial/random perturbations?**
>
> Thank you for the question. While low sensitivity (to token-wise perturbations) correlates with better robustness, they are not equivalent. Specifically, in App. A.2, we evaluate sensitivity to Gaussian noise added across the input and observe that this metric does not distinguish transformers from other architectures as clearly as the sensitivity to token-wise Gaussian perturbations.

---

### Author Response · Authors · 2024-11-22

We sincerely thank the reviewers for their time and effort in reviewing our work. We are encouraged that the reviewers find the topic of understanding transformers important (kdeM, eYcM), the use of sensitivity novel/original (A7Md, kdeM, eYcM), the finding that transformers learn low sensitivity functions interesting (pCqz), the writing clear and easy to follow (kdeM, eYcM). The reviewers appreciate our experiment design (eYcM) as well as the comprehensiveness and consistency of the results (pCqz, A7Md, kdeM), which demonstrate that transformers have lower sensitivity compared to other architectures across both synthetic and realistic vision and language datasets. They also find the theoretical results interesting (kdeM) and the connections between sensitivity and other phenomena like robustness and grokking new (pCqz, A7Md) and important (kdeM).

We have addressed the comments in the responses to each reviewer and will incorporate all feedback into the paper.

---

### Meta-Review · Area_Chair_3znp · 2024-12-20

**Metareview:**

This paper extends the original notion of sensitivity in Boolean function analysis to an empirically measurable quantity and study it on transformers. The main conclusions contain three points, including lower sensitivity correlates well with robustness (this is almost by definition), flat minima, and grokking. The use of sensitivity to understand transformers is not new, e.g., Bhattamishra et al., and the main novel points are its correlations with flat minima and grokking.

During the rebuttal the reviewers have pointed out a few over-claims which have been addressed by the authors. There are other concerns regarding the scale of the experiments that have not been properly addressed. I also agree with Reviewer eYcM that more extensive experiments on larger-scale models/datasets could be helpful to at least showcase the generality of the claims given that the main focus of this paper is on the empirical phenomenon rather than theoretical contributions.

Overall, the empirical connection between sensitivity to flat minima and grokking is interesting and might lead to follow-up work. I recommend acceptance.

**Additional Comments On Reviewer Discussion:**

3 out of 4 reviewers are enthusiastic about the paper, including one of the reviewers who initially rated negatively of the paper. The last reviewer is an expert reviewer whose comments help to better position the contributions of this work. Overall I feel it's an interesting phenomenon worth sharing with the community and can potentially lead to some follow-up work, hence I recommend acceptance, but as a poster rather than spotlight or oral given the limitation on experiments.

---

### Decision · Program_Chairs · 2025-01-22

Accept (Poster)